# HNF4α regulates sulfur amino acid metabolism and confers sensitivity to methionine restriction in liver cancer

Qing Xu[1], Yuanyuan Li[2], Xia Gao [3], Kai Kang [2], Jason G. Williams[4], Lingfeng Tong[5], Juan Liu[3], Ming Ji[1], Leesa J. Deterding[4], Xuemei Tong[5], Jason W. Locasale [3], Leping Li [2], Igor Shats [1✉] & Xiaoling Li [1✉]

Methionine restriction, a dietary regimen that protects against metabolic diseases and aging, represses cancer growth and improves cancer therapy. However, the response of different cancer cells to this nutritional manipulation is highly variable, and the molecular determinants of this heterogeneity remain poorly understood. Here we report that hepatocyte nuclear factor 4α (HNF4α) dictates the sensitivity of liver cancer to methionine restriction. We show that hepatic sulfur amino acid (SAA) metabolism is under transcriptional control of HNF4α. Knocking down HNF4α or SAA enzymes in HNF4α-positive epithelial liver cancer lines impairs SAA metabolism, increases resistance to methionine restriction or sorafenib, promotes epithelial-mesenchymal transition, and induces cell migration. Conversely, genetic or metabolic restoration of the transsulfuration pathway in SAA metabolism significantly alleviates the outcomes induced by HNF4α deficiency in liver cancer cells. Our study identifies HNF4α as a regulator of hepatic SAA metabolism that regulates the sensitivity of liver cancer to methionine restriction.

[1] Signal Transduction Laboratory, National Institute of Environmental Health Sciences, Research Triangle Park, Durham, NC 27709, USA. [2] Biostatistics and Computational Biology Branch, National Institute of Environmental Health Sciences, Research Triangle Park, Durham, NC 27709, USA. [3] Department of Pharmacology and Cancer Biology, Duke University School of Medicine, Durham, NC 27710, USA. [4] Mass Spectrometry Research and Support Group, National Institute of Environmental Health Sciences, Research Triangle Park, Durham, NC 27709, USA. [5] Department of Biochemistry and Molecular Cell Biology, Shanghai Key Laboratory for Tumor Microenvironment and Inflammation, Key Laboratory of Cell Differentiation and Apoptosis of Chinese Ministry of Education, Shanghai Jiao Tong University School of Medicine, 200001 Shanghai, China. ✉email: igor.shats@nih.gov; lix3@niehs.nih.gov

Increasing evidence indicates that availability of dietary nutrients, including amino acids and fatty acids, has profound impacts on tumor metabolism, growth, and therapeutic outcomes[1–7]. For example, restriction of dietary methionine, a sulfur-containing essential amino acid enriched in animal products, has been shown to suppress proliferation and progression of a variety of tumors, including colon, prostate, and breast cancer[8–13]. This dietary intervention can impact metabolic flux in one-carbon metabolism, inhibit tumor growth, and sensitize tumors to chemotherapy and radiotherapy in certain human cancer cells in a tumor-cell autonomous manner[14]. However, different human cancer cells have varying degrees of methionine dependence[15], and the underlying molecular determinants of this heterogeneity are still unclear.

Systemic methionine metabolism, more broadly sulfur amino acid (SAA) metabolism, is thought to mainly take place in the liver, a central metabolic organ that metabolizes half of all dietary methionine[16]. Once transported into the liver, dietary methionine is converted to S-Adenosyl methionine (SAM) primarily through the action of MAT1A, a liver-specific methionine adenosyltransferase. S-adenosylhomocysteine (SAH) generated from SAM via the transmethylation reaction is then hydrolyzed to form homocysteine (Hcy) and remethylated back to methionine through betaine-homocysteine S-methyltransferase (BHMT). Hcy can also enter the transsulfuration pathway to form cystathionine (Ctt) and cysteine (Cys) through cystathionine-beta-synthase (CBS) and cystathionine gamma-lyase (CTH). CBS and CTH also produce hydrogen sulfide ($H_2S$)[17,18], a gasotransmitter whose metabolism and biological functions in cancer, aging, and age-associated diseases are still being unraveled[19,20]. Cys can be further used to produce antioxidant glutathione (GSH), or generate taurine through cysteine dioxygenase (CDO1) (Fig. 1a). Therefore, aside from being indispensable for protein synthesis, methionine plays important roles in tissue and systemic sulfur metabolism, antioxidant defense, epigenetic regulation, and signaling[21].

Interestingly, many enzymes involved in SAA metabolism, including MAT1A, GNMT, BHMT, and CBS, are reported to be downregulated in human liver tumors, particularly hepatocellular carcinoma (HCC), the most common and deadly form of liver cancer. Reduced expression of these enzymes is also associated with more aggressive tumors and poor prognosis[22–25]. Liver cancer is the fourth leading cause of global cancer death and its incidence is rapidly growing in the US[26]. Risks factors for liver cancer include chronic hepatitis B/C infection, cirrhosis linked to alcohol abuse, diabetes, and obesity[27]. Liver cancer is commonly diagnosed at an advanced stage when tumors are already resistant to conventional chemotherapy or radiotherapy. Development of therapeutic agents for liver cancer has been challenging. For decades, sorafenib, a multiple kinase inhibitor, has been the sole approved first-line treatment for advanced HCC despite the fact that it offers only three-month survival benefit over placebo[28]. Therefore, there is an urgent need to develop therapeutic strategies for effective treatment of liver cancer.

The dysregulated SAA metabolism in liver cancer suggests that manipulation of this metabolic pathway through dietary methionine intervention could serve as a promising treatment approach. However, currently the molecular mechanisms underlying aberrant SAA metabolism in liver tumors are poorly understood. Whether methionine restriction can lower the risk of liver cancer and/or increase the sensitivity of liver tumors to available chemotherapy also remains unexplored.

In this study, we establish a link between hepatocyte nuclear factor 4α (HNF4α) and SAA metabolism in liver cancer. HNF4α, the master regulator of hepatic genes, is a member of nuclear receptor family of transcriptional factors that is critical for maintenance of hepatocyte identity and specification of hepatic

functions[29,30]. Downregulated in HCC[31–33], HNF4α is considered as a tumor suppressor that represses the development of HCC and inhibit epithelial-mesenchymal transition (EMT), a process that promotes cancer progression and metastasis[34–36]. Although hepatic HNF4α is known to regulate genes essential for gluconeogenesis, bile acid synthesis, cholesterol and lipid metabolism[36], whether and how HNF4α modulates SAA metabolism are not known. Through bioinformatic analyses, metabolomics, and molecular, cellular and in vivo characterizations, we demonstrate that HNF4α plays a central role in controlling hepatic SAA metabolism and dictating sensitivity to methionine restriction in liver cancer both in vivo and in vitro.

## Results

**HNF4α and SAA enzymes are positively correlated in liver cancer.** It has long been noticed that SAA metabolism is one of the major dysregulated metabolic pathways in liver tumors[22,37]. Our analysis of The Cancer Genome Atlas (TCGA) human Liver Hepatocellular Carcinoma (LIHC) dataset confirmed that key genes involved in SAA metabolism, including MAT1A, BHMT, CBS, CTH, and CDO1, are suppressed in HCC compared to normal liver (Supplementary Fig. 1a). Moreover, HCC patients with low tumor expression of these key SAA enzymes had significantly worse prognosis than those with high expression of these genes (Supplementary Fig. 1b). These data suggest that SAA metabolism is frequently disrupted in liver tumors and this disruption is correlated with patient prognosis.

Notably, HNF4α expression was also progressively reduced in tumors of the HCC patients with the advancement of the tumor stage in the TCGA LIHC dataset (Supplementary Fig. 1c), and lower expression of HNF4α was associated with shorter patient survival (Supplementary Fig. 1d). As in many other cancers, HCC progression is characterized by EMT with loss of epithelial markers (e.g., E-cadherin, ZO-1, Cytokeratin) and gain of mesenchymal markers (e.g., Vimentin, TWIST1, ZEB1, CD44, SNAI1, SNAI2)[38]. HNF4α has been reported to inhibit hepatocarcinogenesis by suppression of EMT[31,34–36]. To understand the molecular mechanisms underlying dysregulation of SAA metabolism during liver cancer development, we first investigated the relationship between the expression levels of five well-established hepatic HNF4α target genes, five general mesenchymal markers, and eight SAA metabolic enzymes in the TCGA LIHC dataset of 373 HCC patients. As shown in Fig. 1b, the mRNA levels of seven out of eight analyzed SAA metabolic enzymes (orange) were clustered together with those of HNF4α and other liver-specific functional genes (red), whereas mesenchymal marker genes (blue) formed a separate cluster. The only exception among eight analyzed SAA enzymes is MAT2A, a ubiquitously expressed methionine adenosyltransferase whose expression is primarily controlled by post-transcriptional RNA methylation that induces efficient splicing and mRNA stabilization in response to methionine starvation[39,40]. Consistent with their clustering patterns, the expression of five key SAA metabolic genes, MAT1A, BHMT, CBS, CTH, and CDO1, was positively correlated with that of HNF4α in both non-viral and viral HCC patients (Fig. 1c, Supplementary Fig. 2a, b). In contrast, their expression was negatively correlated with that of TWIST1, a mesenchymal marker, in these HCC patients (Fig. 1d). Notably, the negative correlation between key SAA enzymes and TWIST1 was in a comparable range as that between HNF4α and TWIST1 (Fig. 1d). These observations raise the possibility that the expression of key SAA metabolic enzymes is under control of HNF4α in human liver tumors.

To further test this possibility, we performed a cluster analysis of RNA-seq data from 25 liver cancer cell lines derived from

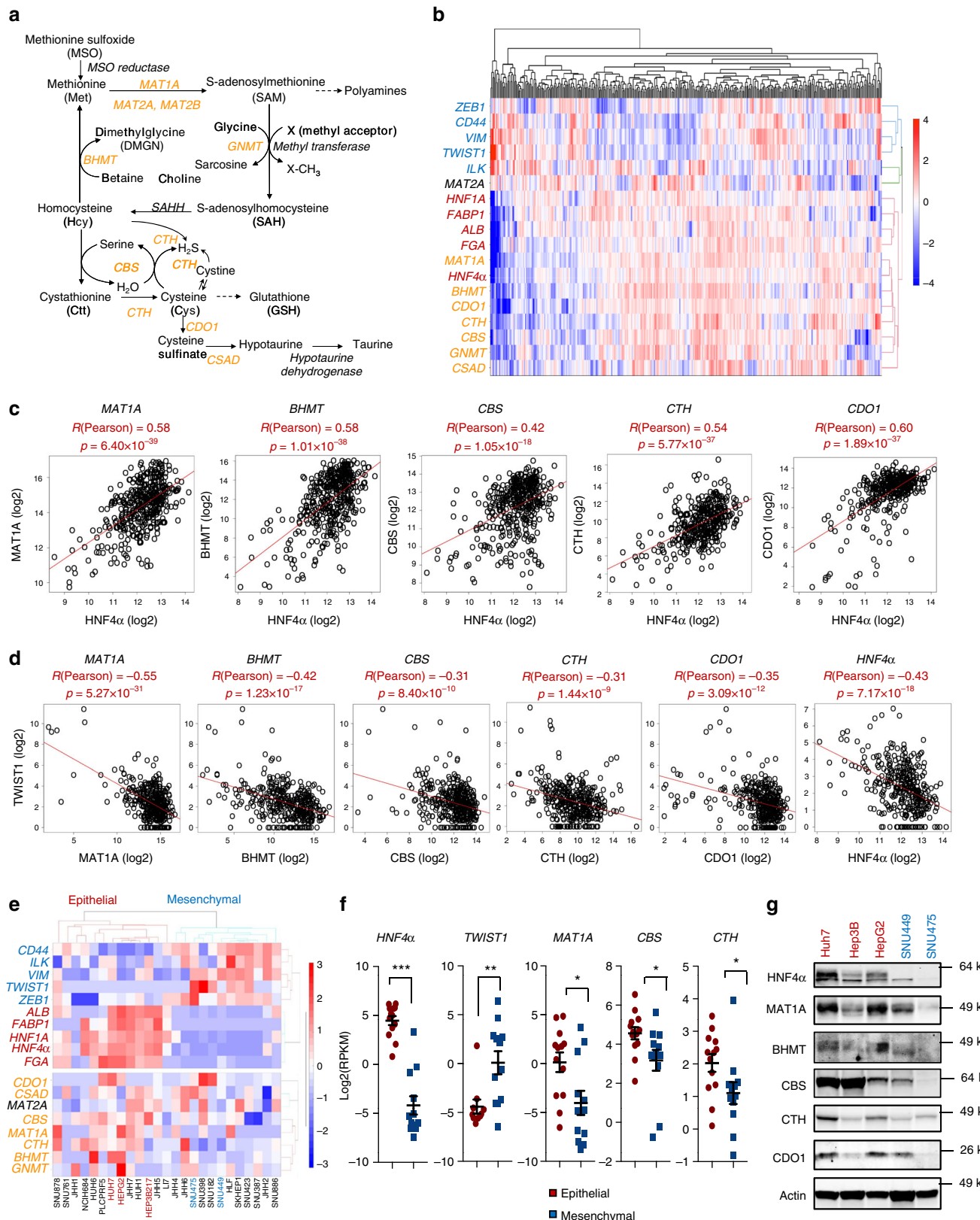

human liver tumors in the Broad Institute Cancer Cell Line Encyclopedia (CCLE) database. Based on their mRNA expression levels of liver-specific markers, including *HNF4α* and its direct target *HNF1*, and mesenchymal markers, these liver cancer cell lines can be clustered into two groups (13 epithelial vs. 12 mesenchymal) (Fig. 1e). In line with our observations in HCC

patients (Fig. 1b), three key SAA metabolic enzymes, *MAT1A*, *CBS* and *CTH*, were significantly enriched in the epithelial group together with *HNF4α* and liver-specific markers (Fig. 1e, f). Additional cluster analyses using RNA-seq data from 81 human liver cancer cell lines in LIMORE database[41] confirmed the significant positive correlation of *MAT1A* and *CBS* with *HNF4α*

**Fig. 1 Key SAA metabolic enzymes and HNF4α are correlated in liver cancers. a** SAA metabolic pathways. MAT1A methionine adenosyltransferase 1A, MAT2A methionine adenosyltransferase 2A, MAT2B methionine adenosyltransferase 2B, GNMT glycine N-methyltransferase, SAHH adenosylhomocysteinase, BHMT betaine-homocysteine S-methyltransferase, CBS cystathionine beta-synthase, CTH/CSE cystathionine gamma-lyase, CDO1 cysteine dioxygenase type 1, CSAD cysteine sulfinic acid decarboxylase. Key SAA enzymes investigated in this study are highlighted in orange. **b** HNF4α is clustered together with key SAA metabolic enzymes in liver cancer patients. The mRNA levels of HNF4α-regulated liver functional genes (Red), mesenchymal markers (Blue), and key SAA metabolic enzymes (Orange) from 373 liver cancer patients from the TCGA LIHC dataset were clustered and represented by the heatmap as described in Methods. **c** Expression of key SAA metabolic enzymes is positively correlated with HNF4α expression in liver cancer patients ($n = 371$). The pair-wise Pearson correlation coefficient and the corresponding $p$-value between two genes were calculated using MATLAB. Two outlier samples in which HNF4α expression levels were more than 3 interquartile ranges (IQRs) below the first quartile among the 373 samples were removed. **d** Expression of key SAA metabolic enzymes is negatively correlated with *TWIST1* expression in liver cancer patients ($n = 373$). **e** HNF4α is clustered together with key SAA metabolic genes in liver cancer cells. The mRNA levels of HNF4α-mediated liver genes (Red), mesenchymal markers (Blue), and SAA metabolic enzymes (Orange) from 25 liver cancer cells from CCLE database were clustered and represented by the heatmap using MATLAB as described in Methods. **f** Expression of *MAT1A*, *CBS* and *CTH* is significantly higher in HNF4α-positive epithelial liver cancer cells than in HNF4α-negative mesenchymal liver cancer cells. The mRNA levels of indicated genes were analyzed using 25 liver cancer cells from the CCLE database ($n = 13$ epithelial, 12 mesenchymal). **g** Protein expression of key SAA metabolic enzymes in 3 selected epithelial and 2 mesenchymal liver cancer cell lines (representative immunoblots are shown from at least three independent experiments). For dot plots in **f**, dots depict individual cell lines, values are expressed as mean ± s.e.m., two-tailed, unpaired, non-parametric Mann-Whitney test, ***$p < 0.001$, **$p < 0.01$, *$p < 0.05$.

and liver-specific markers (Supplementary Fig. 2c, d). Further immuno-blotting analysis indicated that three epithelial cell lines Huh7, Hep3B, and HepG2 that express high levels of HNF4α also displayed high levels of many SAA enzymes compared to two mesenchymal cell lines SNU449 and SNU475 that are negative for HNF4α (Fig. 1g). Therefore, the expression of key SAA metabolic enzymes is positively correlated with that of HNF4α in both liver cancer patients and liver cancer cell lines.

Importantly, the positive correlation between HNF4α and SAA metabolic enzymes had functional consequences in liver cancer cells. An unbiased LC-MS-based metabolomic analysis of the small molecule metabolites in HNF4α-positive HepG2 cells and HNF4α-negative SNU449 cells, two widely used cell lines in the research community of liver cancer, revealed that SNU449 cells are significantly different from HepG2 cells in the abundance of 174 metabolites (Supplementary Table 1, $p < 0.05$, $|FC| > 1.5$). Pathway analysis demonstrated that these metabolites were enriched with metabolites from SAA metabolic pathways, particularly cysteine and methionine metabolism (Fig. 2a). Further targeted LC-MS and biochemical assays confirmed that SNU449 cells accumulated methionine and cysteine, but were depleted of SAM, SAH, Ctt, hypotaurine, GSH, and $H_2S$ (Fig. 2b, c). All of these metabolites were regulated by key SAA enzymes that were reduced in SNU449 cells compared to HepG2 cells (Fig. 2d). Taken together, our observations indicate that HNF4α, SAA metabolism genes, and SAA metabolism are positively linked in human liver tumors and cell lines, and that SAA metabolism is altered in HNF4α-negative mesenchymal liver cancer cell lines. Since mesenchymal lines typically originate from invasive and metastatic tumors, our results suggest that reduction of SAA metabolism genes and the consequent rewiring of SAA metabolism may represent hallmarks of liver cancer progression.

**HNF4α-negative liver cancer lines are resistant to MCR.** To evaluate the possible functional impacts of altered SAA metabolism in mesenchymal liver cancer cell lines, we compared the responses of two mesenchymal liver cancer lines, SNU449 and SNU475, to methionine restriction with those of three epithelial cell lines Huh7, Hep3B, and HepG2.

Our LC-MS analysis indicated that our complete DMEM cell culture medium (CM, DMEM medium plus 10% regular Fetal Bovine Serum (FBS)) contains about 130 μM methionine and 160 μM cystine (Cys-Cys), the oxidized dimer form of cysteine. The commonly used dietary methionine restriction regimen has been shown to extend life span, delay aging, prevent metabolic diseases, reduce cancer growth, and sensitize cancer cells to chemotherapy and radiation in mice[14,42,43]. Since this regimen

restricts methionine in the absence of cystine, we restricted both methionine and cystine in our cell culture DMEM medium by combining methionine/cystine-free DMEM with 10% dialyzed FBS, which resulted in a restricted DMEM medium (MCR) containing 0.12 μM methionine and undetectable levels of cystine. Further LC-MS analysis revealed that both media are able to maintain or even increase their respective methionine/cystine concentrations during a 24-h experimental timeframe, as the concentrations of methionine and cystine were 139 μM and 200 μM, respectively, in CM, and 0.15 μM and 0.11 μM in MCR after 24-h cell culture. This observation is consistent with the notion that small peptides or single amino acids can be derived from proteolysis of large serum proteins[44], or from proteolysis of cell components via autophagy or similar processes during cell culture. Interestingly, cysteine was not detectable in these media. In contrast, the intracellular methionine and cysteine levels were dramatically reduced to ~0.3% and undetectable, respectively, after 1 h of methionine/cystine restriction in both HepG2 and SNU449 cells (Supplementary Fig. 3). methionine/cystine restriction also quickly reduced intracellular glutathione levels (Supplementary Fig. 3).

Upon amino acid deprivation, it is established that uncharged transfer RNAs (tRNAs) triggers an adaptive integrated stress response, termed amino acid response (AAR), which activates the expression of stress-responsive transcription factor ATF4[45,46]. As expected, in three epithelial cell lines, restriction of both methionine and cystine for 6 h significantly increased the expression of *ATF4* and its two target genes involved in the regulation of cell stress and apoptosis, *C/EBP homologous protein* (*CHOP*) and *Asparagine Synthetase* (*ASNS*), as well as *Met tRNA synthetase* (Fig. 3a, Huh7, Hep3B, and HepG2). The mesenchymal SNU449 and SNU475 cells, on the other hand, displayed elevated basal levels of *AFT4* and *CHOP* already in the complete medium and failed to further increase the expression of all tested genes upon methionine/cystine restriction (Fig. 3a, SNU449 and SNU475). This finding suggests that mesenchymal cells with dysregulated SAA metabolism are under stress already in regular growth conditions, and are not responsive to cellular stress induced by methionine/cystine restriction. In line with this notion, mesenchymal SNU449 and SNU475 cells were more resistant to cell death caused by a 24-h methionine/cystine restriction compared to epithelial Huh7, Hep3B, and HepG2 cells (Fig. 3b, c). Intriguingly, this mesenchymal resistance was specific to the restriction of methionine/cystine, and not to the depletion of other non-SAA amino acids including leucine (essential), threonine (essential), or glutamine (conditionally essential) (Fig. 3d). This observation suggests that differential responses

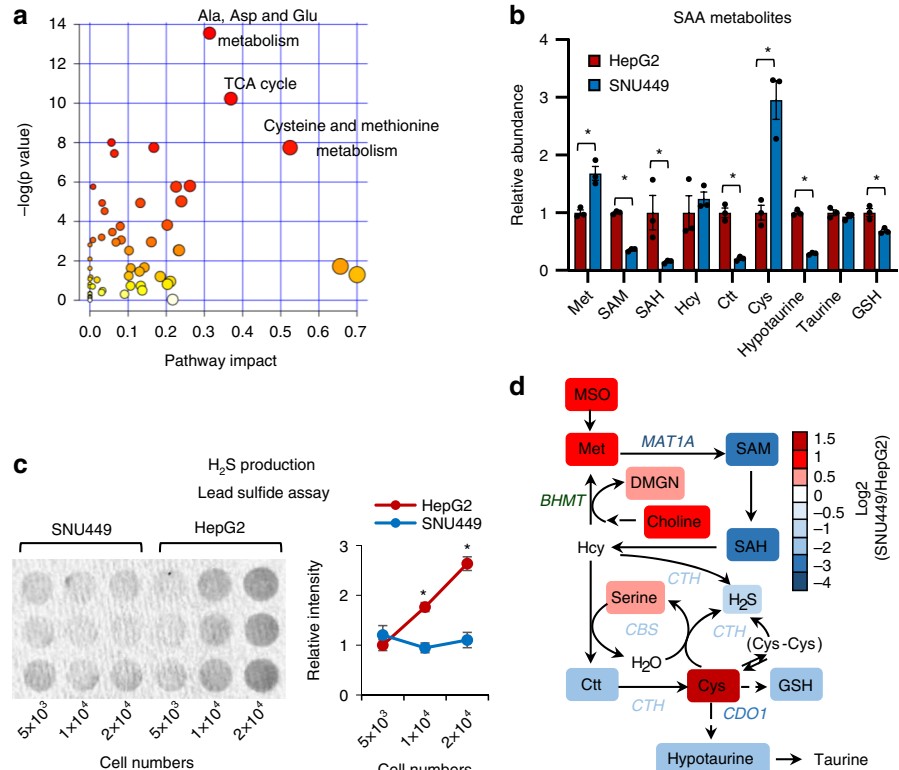

**Fig. 2 HNF4α-deficient mesenchymal liver cancer cells have altered SAA metabolism. a** Metabolite sets involved in SAA metabolism are enriched among the metabolites with differential abundance between HNF4α-negative mesenchymal SNU449 cells and HNF4α-positive epithelial HepG2 cells. The 174 metabolites that displayed significantly different abundances between SNU449 and HepG2 cells were subjected for the pathway enrichment analysis (y axis, enrichment p values) and the pathway topology analysis (x axis, pathway impact values, indicative of the centrality and enrichment of a pathway) in the Pathway Analysis module of MetaboAnalyst 4.0 ($n = 3$ replicates per group, $p < 0.05$, $|FC| > 1.5$). The color of a circle is indicative of the level of enrichment significance, with yellow being low and red being high. The size of a circle is proportional to the pathway impact value of the pathway. **b** SNU449 cells have altered SAA metabolism compared to HepG2 cells. Indicated metabolites were quantified by LC-MS ($n = 3$ replicates per group). **c** SNU449 cells have reduced $H_2S$ production compared to HepG2 cells. The production of $H_2S$ by the indicated number of HepG2 and SNU449 cells was analyzed over 24 h using the lead sulfide assay as described in Methods ($n = 3$ replicates per group). **d** SAA metabolic pathway in SNU449 cells relative to HepG2 cells. The log2 ratios of the relative abundance of metabolites and enzymes in indicated pathways in SNU449/HepG2 cells were presented by color scale ($n = 3$ replicates per group, all colored metabolites were significantly changed in SNU449 cells compared to HepG2 cells with $p < 0.05$). For graphs in (**b**, **c**), values are expressed as mean ± s.e.m., two-tailed, unpaired Student's t-test, $*p < 0.05$.

of epithelial and mesenchymal liver cancer cells to methionine/cystine restriction are not simply because methionine is essential and indispensable for protein synthesis.

Intracellular cysteine depletion can also be induced by sorafenib, the sole approved first-line drug for advanced HCC that inhibits the cystine-glutamate antiporter (system $x_c^-$) in addition to multiple kinases[47,48]. Notably, compared to epithelial liver cancer lines, HNF4α-negative mesenchymal liver cancer lines also showed increased resistance to cell death induced by sorafenib when cultured in complete medium, and methionine/cystine restriction failed to enhance the effect of sorafenib in these cells (Fig. 3e, SNU449 and SNU475). In contrast, HNF4α-positive epithelial liver cancer lines were sensitive to sorafenib treatment when cultured in complete medium, and this sensitivity was further augmented by lowering medium concentrations of methionine/cystine (Fig. 3e, Huh7, Hep3B, and HepG2). Collectively, our results suggest that the status of HNF4α and SAA metabolism may dictate the sensitivity of liver cancer cells to methionine/cystine restriction and sorafenib treatment.

**HNF4α regulates the transcription of SAA metabolic enzymes.** To further understand how HNF4α regulates SAA metabolism and the resulting cellular sensitivity to methionine/cystine

restriction and sorafenib, we tested whether SAA metabolic enzymes are direct transcription targets of HNF4α since the expression of HNF4α is strongly correlated with those of SAA metabolic genes in liver cancer (Fig. 1 and Supplementary Fig. 1). In support of this notion, ENCODE analysis of published ChIP-seq datasets revealed that the promoters of MAT1A, BHMT, and CBS are all bound by HNF4α at multiple sites in HepG2 cells and in human liver (Supplementary Fig. 4a–c). CDO1 has also been previously reported as an HNF4α target gene[49].

We confirmed that HNF4α was significantly enriched on the HNF4α binding sites of these SAA metabolism genes in HepG2 but not HNF4α-negative SNU449 cells by Chromatin Immuno-precipitation (ChIP)-qPCR assay (Fig. 4a). Consistently, knocking down HNF4α by siRNAs significantly reduced the mRNA and/or protein levels of these key SAA metabolic enzymes in HNF4α-positive HepG2, Huh7, and HepB3 cells (Fig. 4b, c and Supplementary Fig. 4d), and normal human hepatocytes (Fig. 4d). Conversely, overexpression of HNF4α in HNF4α-negative SNU449 cells strongly induced the luciferase activities from reporters containing the consensus (WT) HNF4α binding sites of MAT1A, BHMT, or CBS promoters, but this induction was significantly reduced when the binding sites were mutated (Mut) (Supplementary Fig. 4e and Fig. 4e). These results confirmed that

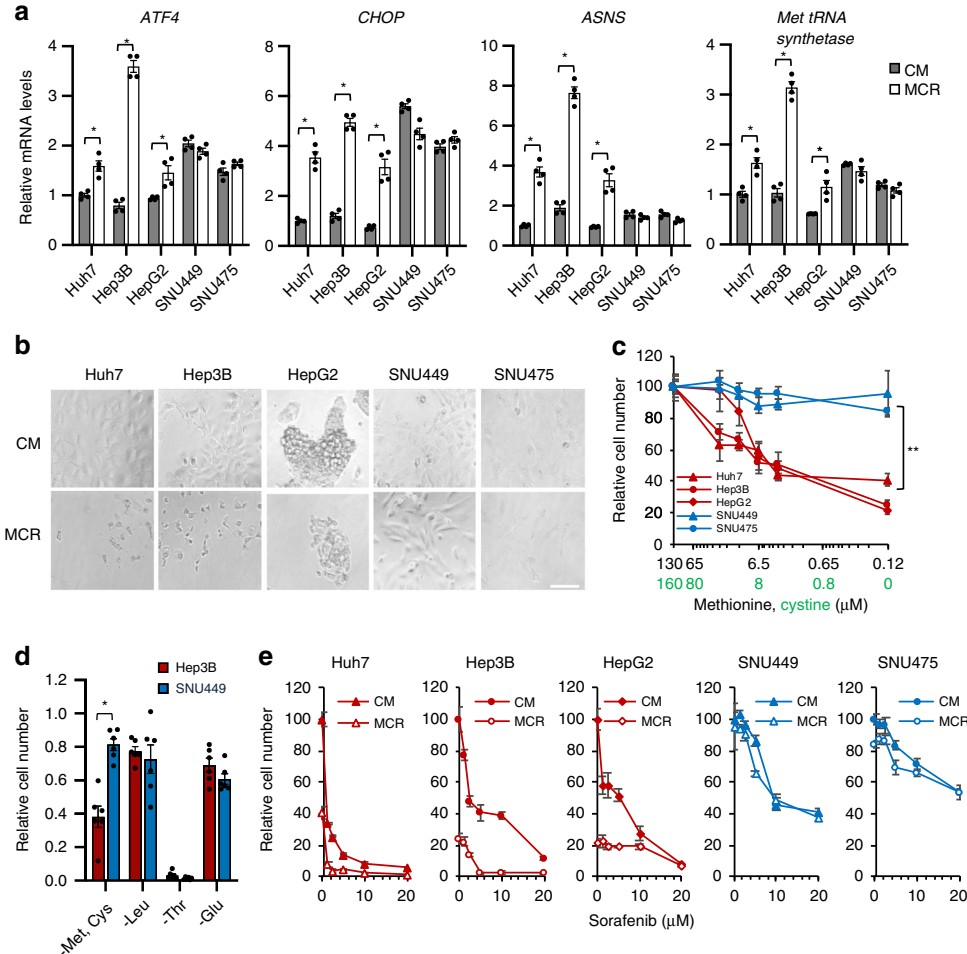

**Fig. 3 HNF4α deficient mesenchymal liver cancer cells are resistant to methionine/cystine restriction-induced and sorafenib-induced cell death. a** Mesenchymal liver cancer cells are resistant to methionine/cystine restriction-induced transcriptional stress response. Five indicated liver cancer cells were cultured in complete medium (CM) or methionine/cystine-restricted medium (MCR) for 6 h. The expression of indicated genes was analyzed by qPCR ($n = 4$ replicates per group). **b** Mesenchymal liver cancer cells are resistant to methionine/cystine restriction-induced cell death. Five indicated liver cancer cells were cultured in complete medium (CM) or Met and Cys restricted medium (MR) for 24 h and analyzed by microscopy (representative images were shown from at least three independent experiments). Bar, 100 μm. **c** Reducing methionine/cystine decreases cell survival in HNF4α-positive epithelial cells but not in HNF4α-negative mesenchymal liver cancer cells. Five liver cancer cells were cultured in medium containing the indicated concentrations of methionine and cystine for 24 h. The relative number of surviving cells was measured by the WST-1 assay ($n = 5$ replicates for each line; ** $p < 0.01$ between slopes of any HNF4α-positive epithelial cells vs. any HNF4α-negative mesenchymal cells using the semilog line equation in the nonlinear fit regression module of Prism8). **d** Mesenchymal SNU449 cells are specifically resistant to methionine/cystine-restriction induced cell death. Epithelial Hep3B cells and mesenchymal SNU449 cells were cultured in complete medium or medium depleted of the indicated exogenous amino acid for 24 h. The relative number of surviving cells was measured by the WST-1 assay ($n = 4$ replicates per group). **e** Methionine/cystine restriction sensitizes epithelial but not mesenchymal liver cancer cells to sorafenib-induced cell death. Five indicated liver cancer cell lines were cultured in complete medium (CM) or methionine/cystine-restricted medium (MCR), together with the indicated concentrations of sorafenib for 24 h ($n = 5$ replicates per group). The relative number of surviving cells was measured by WST-1 assay. For graphs in (**a, c, d, e**), values are expressed as mean ± s.e.m., two-tailed, unpaired Student's $t$-test, * $p < 0.05$.

HNF4α directly controls the transcription of these SAA metabolic enzymes in part through the identified consensus HNF4α binding sites on their promoters. Overexpression of HNF4α in SNU449 cells failed to rescue the expression of endogenous SAA metabolic enzymes (Fig. 4f, top), likely due to the reported epigenetic silencing of these genes in cancer[50–52]. In support of this idea, co-treatment with a histone deacetylase inhibitor trichostatin A (TSA) enabled ectopic HNF4α to promote the expression of these endogenous SAA genes (Fig. 4f, bottom). Together, our data indicate that HNF4α controls SAA metabolism in liver cancer cells through transcriptional regulation of the expression of SAA enzymes.

**HNF4α deficiency alters SAA metabolism and MCR resistance.** To further test the importance of SAA metabolism in HNF4α-mediated metabolic program, we knocked down HNF4α in HepG2 cells and compared their metabolomic profiles with those from HepG2 cells transfected with a control siRNA. Notably, unbiased pathway analysis revealed that the two most significantly altered metabolic pathways in HNF4α-depleted HepG2 cells were two SAA metabolism pathways, taurine/hypotaurine metabolism and cysteine/methionine metabolism (Fig. 5a). Specifically, HNF4α-depleted HepG2 cells showed increased levels of methionine and cysteine but reduced Ctt, hypotaurine, taurine, and $H_2S$ (Fig. 5b–d). This dysregulated SAA metabolism was strikingly similar to that

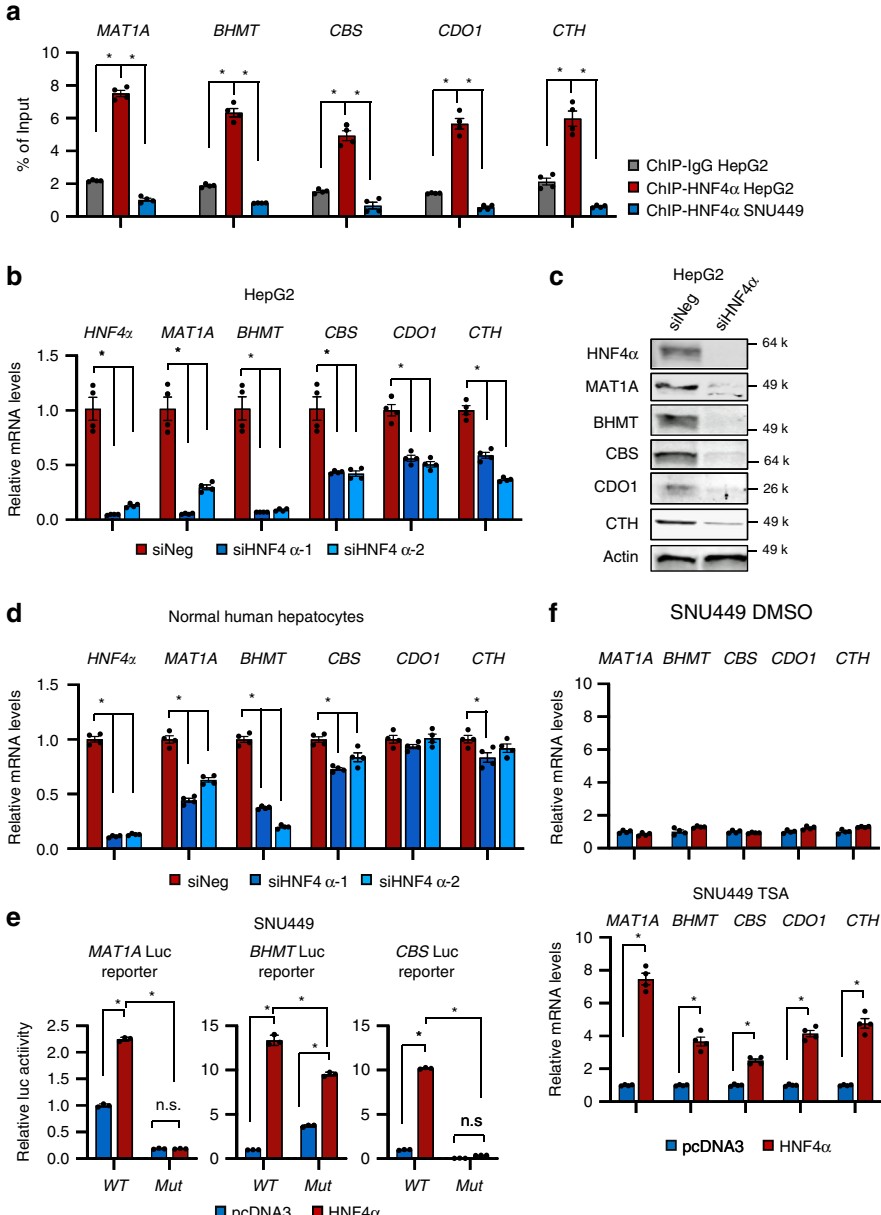

**Fig. 4 HNF4α regulates the expression of SAA metabolic enzymes in liver cells. a** HNF4α binds to the promoters of SAA metabolic genes. HepG2 cells and SNU449 cells were analyzed by ChIP-qPCR for binding of HNF4α to HNF4α binding sites on the promoters of indicated SAA metabolic genes ($n = 4$ replicates per group). IgG ChIP in HepG2 cells and anti-HNF4α ChIP in HNF4α-negative SNU449 cells serve as negative controls. **b** Knockdown of HNF4α in HepG2 cells reduces the mRNA levels of key SAA metabolic enzymes. HepG2 cells were transfected with control siRNA (siNeg) or two independent siRNAs targeting HNF4α (siHNF4α) for 48 h. mRNA levels of the indicated SAA genes were analyzed by qPCR ($n = 4$ replicates per group). **c** HNF4α depletion reduces the protein levels of key SAA metabolic enzymes. HepG2 cells were transfected with control siRNA (siNeg) or a siRNA targeting HNF4α (siHNF4α) for 48 h (representative immunoblots are shown from at least three independent experiments). **d** Knockdown of HNF4α in normal human hepatocytes reduces the mRNA levels of key SAA metabolic enzymes. Normal human hepatocytes were treated and analyzed as in **b** ($n = 4$ replicates per group). **e** Luciferase reporters with mutated HNF4α binding sites from *MAT1A*, *BHMT*, and *CBS* promoters have reduced transactivation in response to HNF4α overexpression in SNU449 cells. HNF4α-negative SNU449 cells were transfected with a control vector (pcDNA3) or a pcDNA3-Flag-HNF4α construct, together with indicated wild type (WT) or mutant (Mut) luciferase reporters. The luciferase activities were measured as described in Methods ($n = 3$ replicates per group). **f** Overexpression of HNF4α induces the expression of key SAA metabolic genes in SNU449 cells after treatment with TSA. HNF4α-negative SNU449 cells transfected with a control vector (pcDNA3) or a pcDNA3-Flag-HNF4α construct were treated with 0.5 µM TSA for 2 days, and the expression of the indicated SAA genes was analyzed by qPCR ($n = 4$ replicates per group). For bar graphs in (**a**, **b**, **d**, **e**, **f**), values are expressed as mean ± s.e.m., two-tailed, unpaired Student's *t*-test, *$p < 0.05$.

observed in mesenchymal SNU449 cells (Fig. 2). As HNF4α suppresses EMT[31,34–36], the shared metabolic changes between above two cell types suggest that defective SAA metabolism driven by HNF4α deficiency could be a feature of EMT, which is commonly associated with tumor metastasis and drug resistance[53].

To investigate the contribution of HNF4α-mediated SAA metabolism to metabolic remodeling during EMT, we compared the global metabolic profiles of epithelial HepG2 cells, HNF4α-depleted HepG2 cells, and mesenchymal SNU449 cells. As shown in Fig. 5e, loss of HNF4α in HepG2 cells significantly changed the

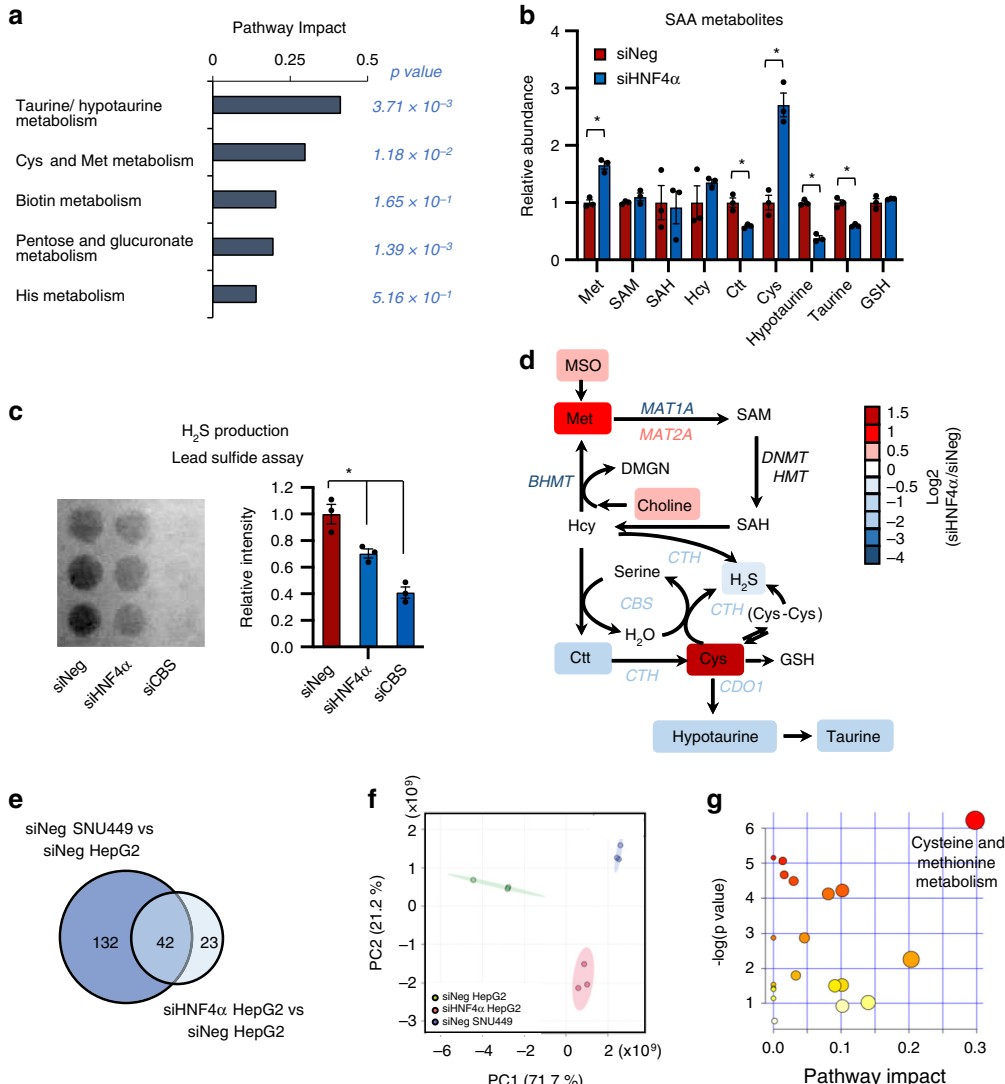

**Fig. 5 Altered SAA metabolism is a common feature of HNF4α defective and mesenchymal liver cancer cells. a** Metabolite sets related to SAA metabolism are enriched in HNF4α-depleted HepG2 cells. The 65 metabolites significantly altered by HNF4α knockdown in HepG2 cells were analyzed in the Pathway Analysis module of MetaboAnalyst 4.0 as described in Methods ($n = 3$ replicates per group, $p < 0.05$, $|FC| > 1.5$). **b** HNF4α depletion alters SAA metabolism. Indicated metabolites were quantified by LC-MS ($n = 3$ replicates per group). **c** HNF4α depletion reduces $H_2S$ production in HepG2 cells. The production of $H_2S$ from indicated cells were analyzed over 24 h using the lead sulfide assay as described in Methods. HepG2 cells with knockdown of CBS, a key $H_2S$ producing enzyme, were used as a negative control ($n = 3$ replicates per group). **d** SAA metabolic pathway in siNeg cells vs. siHNF4α HepG2 cells. The log2 ratios of the relative abundance of metabolites and enzymes in the indicated pathways in siHNF4α/siNeg HepG2 cells are presented by a color scale ($n = 3$ replicates per group, all colored metabolites were significantly changed with $p < 0.05$). **e** HNF4α-depleted HepG2 cells and SNU449 cells have significantly overlapping metabolic profiles (hypergeometric $p < 0.05$). The indicated significantly altered metabolites between siNeg SNU449 vs. siNeg HepG2 and siHNF4α vs. siNeg HepG2) are visualized by a Venn-diagram ($p < 0.05$, $|FC| > 1.5$). **f** HNF4α depletion shifts HepG2 cells metabolically toward SNU449 cells. All detectable metabolites in siNeg HepG2 cells (green), siHNF4α HepG2 cells (red), and siNeg SNU449 cells (blue) were analyzed by the Principal Component Analysis (PCA). **g** SAA metabolites are enriched among the commonly altered 34 metabolites in HNF4α-depleted HepG2 cells and SNU449 cells. The 34 metabolites that were altered in the same direction in SNU449 cell and siHNF4α HepG2 were analyzed in the Pathway Analysis module of MetaboAnalyst 4.0 ($n = 3$ replicates per group, $p < 0.05$, $|FC| > 1.5$). The color of a circle is indicative of the level of enrichment significance, with yellow being low and red being high. The size of a circle is proportional to the pathway impact value of the pathway. For graphs in (**b**, **c**), values are expressed as mean ± s.e.m., two-tailed, unpaired Student's $t$-test, *$p < 0.05$.

abundance of 65 metabolites (Supplementary Table 2, $p < 0.05$, $|FC| > 1.5$), whereas the abundances of 174 metabolites were significantly different between control HepG2 and SNU449 cells (Supplementary Table 1). Among these, 42 metabolites were shared between HNF4α-depleted HepG2 cells and SNU449 cells (Supplementary Table 3). In a Principal Component Analysis (PCA) of all detectable metabolites, knocking down HNF4α in HepG2 cells shifted the global metabolic profile toward that of SNU449 cells at the PC1 axis compared to control HepG2 cells

(Fig. 5f). Therefore, HNF4α deficiency in epithelial liver cancer cells results in a global metabolic shift towards mesenchymal liver cancer cells. HNF4α-depleted HepG2 cells also metabolically clustered together with SNU449 cells but not with control HepG2 in a cluster analysis of all significantly changed metabolites in HNF4α-depleted HepG2 cells (Supplementary Fig. 5 and Supplementary Table 2). Remarkably, metabolites involved in SAA metabolism were the most significantly enriched metabolites among the 34 metabolites that were significantly changed in the

same direction in both HNF4α-depleted HepG2 cells and SNU449 cells (Fig. 5g and Supplementary Table 3), suggesting that impaired SAA metabolism is one of the major shared metabolic characteristics of HNF4-depleted epithelial cells and HNF4-negative mesenchymal liver cancer cells.

In addition to similar metabolic defects in SAA metabolism, both HNF4α-depleted HepG2 cells and SNU449 cells had significantly increased intracellular levels of reactive oxygen species (ROS) (Supplementary Fig. 6a) compared to control HepG2 cells when cultured in the complete medium. Methionine/cystine restriction increased ROS levels in control HepG2 cells but not further in HNF4α-depleted HepG2 cells and SNU449 cells (Supplementary Fig. 6a and 6b, MCR vs. CM), suggesting that HNF4α-depleted cells with dysregulated SAA metabolism already experience elevated oxidative stress under normal condition and are nonresponsive to further cellular stress induced by methionine/cystine restriction. On the other hand, although both HNF4α-depleted HepG2 cells and SNU449 cells displayed reduced proliferation, as evident by a reduced fraction of cells in S-phase compared to control HepG2 cells (Supplementary Fig. 6c, siHNF4α vs. siNeg HepG2 cells, siNeg SNU449 vs. siNeg HepG2 cells), methionine/cystine restriction was able to further reduce the fraction of S-phase cells while increasing G1-phase cells in these cells (Supplementary Fig. 6c, MCR vs. CM), indicating that methionine/cystine restriction induces G1 arrest independent of the cellular status of HNF4α. Consistent with the cell cycle and proliferation results, both HNF4α-depleted HepG2 cells and SNU449 cells displayed reduced protein synthesis compared to control HepG2 cells, with reduced intensities of puromycin-labeled peptides based on the SUnSET assay[54] (Supplementary Fig. 6d), but methionine/cystine restriction suppressed protein synthesis in all these cells regardless of their HNF4α status (Supplementary Fig. 6d).

To directly test the role of HNF4α in SAA metabolism and cellular stress resistance, we investigated the impact of HNF4α deficiency on EMT and its associated resistance to methionine/cystine restriction and sorafenib. Consistent with previous reports[31,34–36], knocking down HNF4α in epithelial HepG2 cells resulted in a decreased expression of epithelial markers E-cadherin (CDH1) and CPED1 yet an increased level of a number of mesenchymal markers[55] (Fig. 6a), along with enhanced mesenchymal cellular morphology (Fig. 6b) and massively increased cell migration in a transwell assay (Fig. 6c). Knocking down HNF4α also reduced the expression of epithelial markers and/or increased the levels of mesenchymal markers in epithelial Huh7 cells and in normal human hepatocytes (Supplementary Fig. 7a). Notably, knocking down HNF4α in three epithelial liver cancer cell lines increased resistance to cell death induced by methionine/cystine restriction (reduced apoptosis in Fig. 6d, and increased surviving cell number in 6e and Supplementary Fig. 7b, siHNF4α vs. siNeg) or sorafenib treatment (Fig. 6f, g, and Supplementary Fig. 7b, siHNF4α Sorafenib vs. siNeg Sorafenib).

To further confirm above observations, we investigated whether HNF4α-depleted HepG2 cells also display enhanced resistance to methionine/cystine restriction in vivo. Since HNF4α is required for maintenance of hepatocyte identity[29,30], we chose to use siRNA-treated HepG2 cells for an in vivo xenograft experiment based on a xenograft mouse model established to generate HNF4α-deficient tumors[56]. In this study, the authors demonstrated that siRNA-mediated knockdown of HNF4α initiates a microRNA-inflammatory feedback loop that continuously suppresses HNF4α expression and sustains a stable phenotype of tumorigenesis[56]. Specifically, we injected control (siNeg) and HNF4α knockdown (siHNF4α) HepG2 cells into nude mice. When the average tumor volume reached 200 mm³, we randomized these mice into two groups and fed them either a

methionine-restricted diet containing 0.172% DL-methionine and no cystine (MCR) or a control diet containing 0.86% DL-methionine with no cystine (CTR). A similar MCR diet has been recently shown to reduce plasma methionine levels by 50% within two days and alter methionine metabolism in colorectal PDX tumors and liver tissues in mice[14]. We confirmed that the MCR diet that contains 0.172% DL-methionine and no cystine suppresses liver cancer growth in a diethylnitrosamine (DEN)/high-fat-diet (HFD)-induced liver cancer model in mice compared to the CTR diet (Supplementary Fig. 8). Xenografted tumors from siHNF4α HepG2 cells maintained partial HNF4α knockdown during our experiment timeframe (about 5 weeks, Supplementary Fig. 9a). Feeding mice with the MCR diet significantly inhibited the growth of xenografted control HepG2 tumors with enhanced tumor damage and death in vivo (Fig. 6h and Supplementary Fig. 9b, siNeg MCR vs. siNeg CTR), whereas knocking down HNF4α significantly blunted the inhibitory impact of the MCR diet on the growth and proliferation of HepG2 tumors (Fig. 6h, i and Supplementary Fig. 6d, siHNF4α MCR vs. siNeg MCR). Therefore, the status of HNF4α affects the sensitivity of liver tumors to methionine restriction both in vitro and in vivo.

**Defective transsulfuration is partially responsible for EMT.** To further assess the importance of SAA metabolism in HNF4α-mediated tumor suppression and stress sensitivity in liver cancer, we knocked down key SAA metabolic enzymes individually in epithelial HepG2 cells (Supplementary Fig. 10a) and analyzed whether deficiency of any of these enzymes mimics HNF4α deficiency-induced mesenchymal characteristics. Consistent with observations in Fig. 5, knocking down HNF4α in HepG2 cells led to significantly increased resistance to stress induced by methionine/cystine restriction or sorafenib, as indicated by the reduced activation of caspase (Fig. 7a, top panels, siHNF4α vs. siNeg), reduced induction of CHOP, a proapoptotic ATF4 target gene (Fig. 7a, bottom panels, siHNF4α vs. siNeg), and enhanced cell survival (Fig. 7b, siHNF4α vs. siNeg). Similar to HNF4α knockdown, knocking individual SAA enzymes led to various degrees of resistance to methionine/cystine restriction or sorafenib treatment (Fig. 7a, b), various degrees of morphological alterations (Supplementary Fig. 10b), and increased cell migration (Fig. 7c). Particularly, depletion of CBS or CDO1 increased stress resistance, and induced cell morphological changes and cell migration to a degree comparable to those induced by HNF4α knockdown (Fig. 7a, b, and Supplementary Fig. 10b). CBS is a key enzyme in the transsulfuration pathway mediating Ctt and H₂S production, and CDO1 is a critical enzyme in taurine synthesis, as well as maintenance of the hepatic intracellular free cysteine range. These observations suggest that a defective transsulfuration pathway with reduced production of Ctt, H₂S, or taurine, may be sufficient to recapitulate the outcomes induced by HNF4α deficiency in epithelial liver cancer cells. In support of this idea, incubation with a H₂S donor NaSH partially prevented the resistance of HepG2 cells to methionine/cystine restriction upon HNF4α knockdown (Fig. 7d, middle). Supplementation of Ctt, NaSH, or taurine also significantly alleviated the resistance of HepG2 to the sorafenib treatment in response to HNF4α knockdown (Fig. 7d, right), and attenuated the cell migration induced by HNF4α deficiency (Fig. 7e). Interestingly, addition of cystine, a major SAA that directly participates in the transsulfuration reactions (Fig. 1a), partially rescued the methionine/cystine restriction-reduced cell survival in control HepG2 cells but not in HNF4α-depleted HepG2 nor HNF4α-negative SNU449 cells (Fig. 7f), suggesting that HNF4α-regulated cysteine metabolism significantly contributes to cellular sensitivity to MCR.

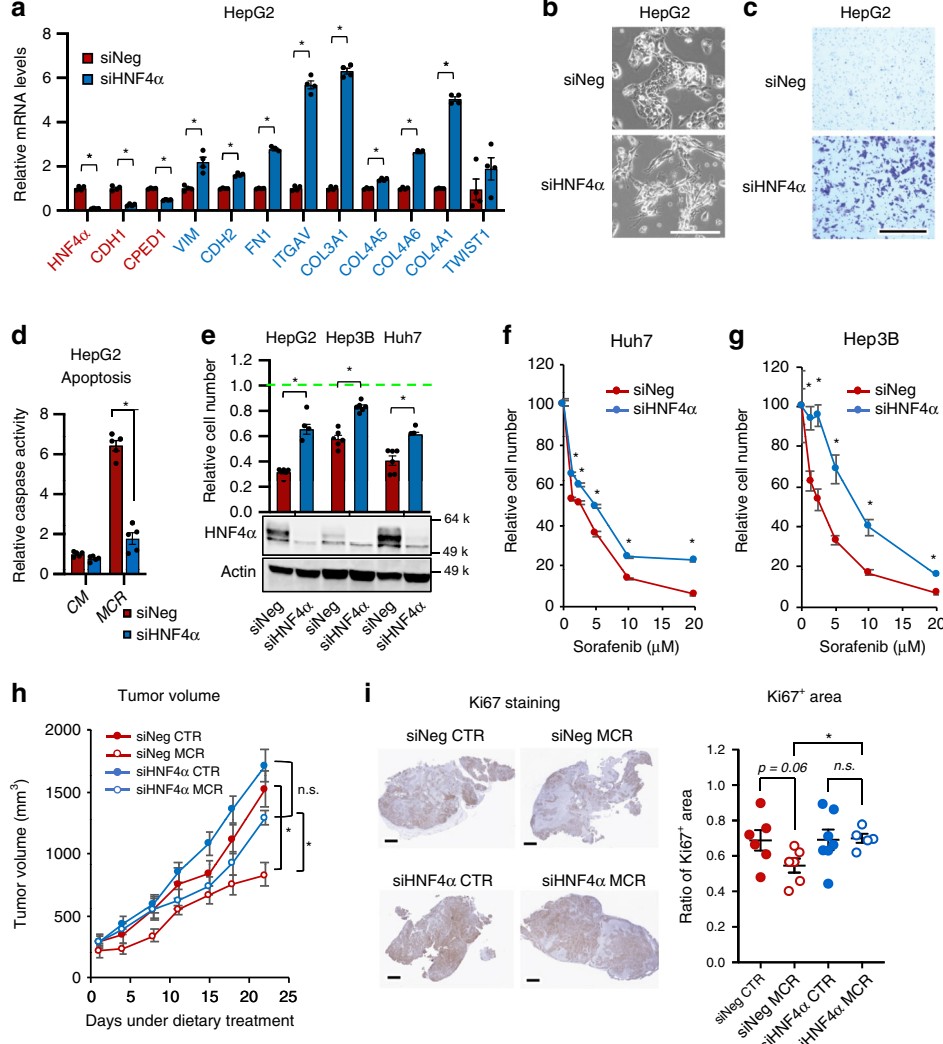

**Fig. 6 HNF4α depletion induces EMT and resistance to sulfur amino acid restriction and sorafenib. a** HNF4α depletion represses epithelial markers while inducing mesenchymal markers. The expression of indicated epithelial (Red) and mesenchymal (Blue) markers was analyzed by qPCR ($n = 4$ replicates per group). **b** HNF4α depletion induces mesenchymal morphology. Bar, 200 μm. **c** HNF4α depletion enhances cell migration. SiNeg and siHNF4α HepG2 cells were subjected to a transwell assay as described in Methods. Bar, 500 μm. **d** HNF4α depletion confers resistance to MCR induced apoptosis. Indicated cells were cultured in CM or MCR medium for 24 h, then analyzed by Caspase Activity Assay ($n = 5$ replicates per group). **e** HNF4α depletion in liver cancer cells induces resistance to MCR. Indicated cells were cultured in CM or MCR medium for 24 h ($n = 5$ replicates per group). The WST-1 reading in CM was normalized to 1 for each cell line (Green dotted line). **f–g** HNF4α depletion confers resistance to sorafenib-induced cell death in Huh7 and HepB3 cells. Indicated cells were cultured in CM containing the indicated concentrations of sorafenib for 24 h ($n = 5$ replicates per group). **h** HNF4α-depleted HepG2 xenograft tumors are resistant to methionine restriction-induced growth inhibition. Mice bearing siNeg and siHNF4α HepG2 xenografted tumors were fed with a control diet (CTR) or methionine-restricted diet (MCR), and tumor volumes were monitored ($n = 6$ tumors in siNeg CTR, 6 tumors in Neg MCR, 7 tumors in siHNF4α CTR, and 5 tumors in siHNF4α MCR). **i** HNF4α-depleted HepG2 xenografted tumors are resistant to methionine restriction-induced repression of Ki67. Ki67 in xenografted tumors was immuno-stained and quantified as described in Methods ($n = 6$ tumors in siNeg CTR, 6 tumors in Neg MCR, 7 tumors in siHNF4α CTR, and 5 tumors in siHNF4α MCR). Representative IHC images are shown, bars: 1 mm. For images in (**b**, **c**), representative images are shown from at least three independent experiments. For graphs in (**a**, **d**, **e**, **f**, **g**), values are expressed as mean ± s.e.m., two-tailed, unpaired Student's t-test, *$p < 0.05$. For plots in (**h**, **i**), values are expressed as mean ± s.e.m., two-tailed, unpaired, non-parametric Mann–Whitney test, *$p < 0.05$. For the dot plot in (**i**), dots depict individual mice.

Further gene expression analyses revealed that at the transcriptional level, knocking down individual SAA enzymes has distinct impacts on EMT changes that are induced by HNF4α deficiency (Fig. 6a). As shown in Fig. 8a and Supplementary Fig. 10a, knocking down CBS in HepG2 cells primarily elevated the expression of a number of mesenchymal markers (e.g., *COL3A1*) that were also induced in HNF4α-depleted HepG2 cells, whereas knocking down other SAA enzymes modestly reduced the expression of two epithelial markers that were repressed in HNF4α depleted cells. Supplementation of CBS products, Ctt and particularly NaSH, also

significantly repressed the expression of a number of HNF4α deficiency-induced mesenchymal markers (Fig. 8b). Furthermore, knocking down CBS resulted in a 10-fold induction of a master regulator of EMT, SNAI2 (SLUG) (Supplementary Fig. 10c). These observations, together with the finding that Ctt and NaSH strongly repressed cell migration in HNF4α-depleted cells (Fig. 7e), raised the possibility that CBS may be the major effector in HNF4α-mediated EMT suppression. In agreement with this possibility, overexpression of CBS alone significantly repressed HNF4α deficiency-induced cell migration in HepG2 cells (Fig. 8c).

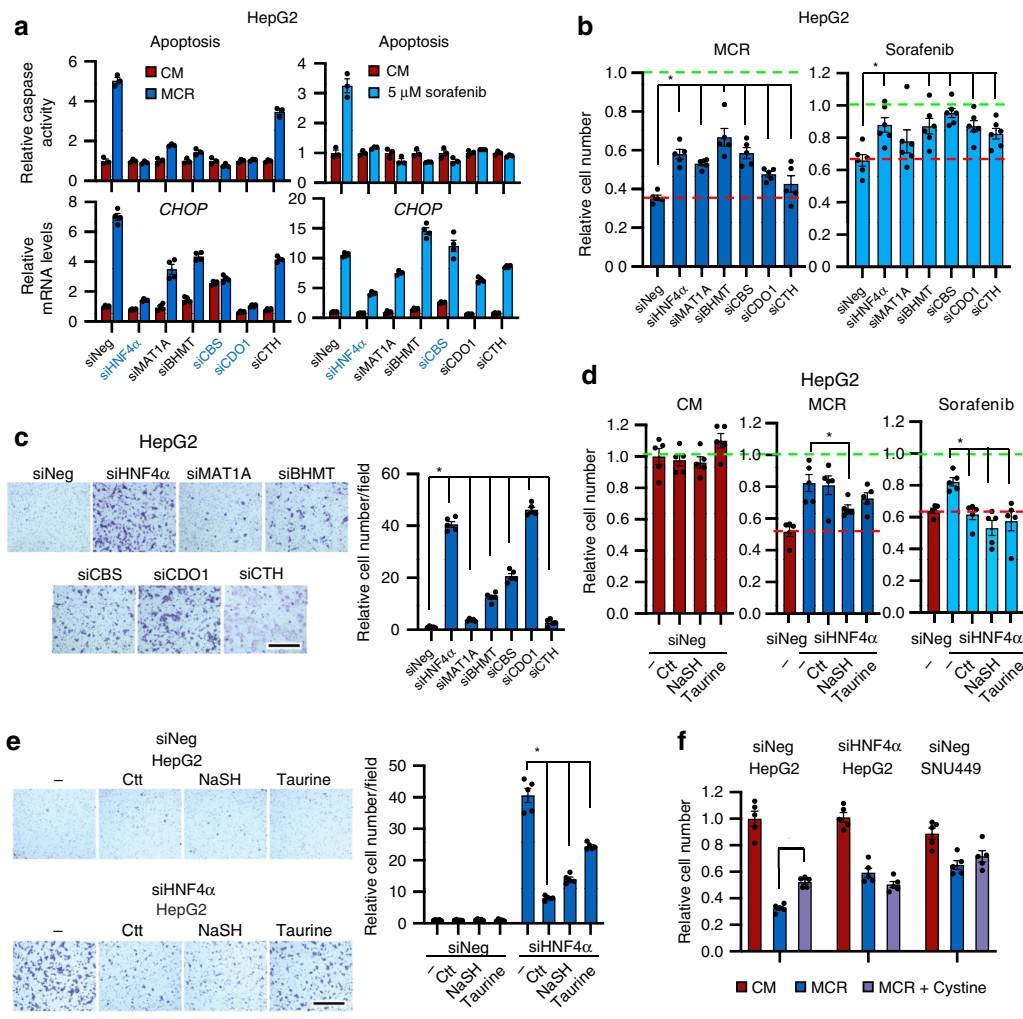

**Fig. 7 Transsulfuration metabolic genes affect stress sensitivity and the migration ability of liver cancer cells. a** Knocking down SAA metabolic genes or HNF4α in HepG2 cells confers resistance to MCR-induced and sorafenib-induced stress. Indicated cells were cultured in CM or MCR medium (left) or in CM containing 5 μM sorafenib (right) for 24 h. Apoptosis was analyzed by a caspase activity assay ($n = 3$ replicates per group) and the expression of *CHOP* was analyzed by qPCR ($n = 4$ replicates per group). **b** Knockdown of SAA metabolic genes or HNF4α in HepG2 cells confers resistance to MCR-induced or sorafenib-induced cell death. Indicated cells were treated as in **a** ($n = 5$ replicates per group for MCR and $n = 6$ replicates per group for sorafenib). The cell number in CM was normalized to 1 (green dotted line), and the cell survival of MCR siNeg cells was denoted by a red dotted line. **c** Knockdown of SAA metabolic genes or HNF4α in HepG2 cells enhances cell migration. Indicated cells were subjected to a transwell assay as described in Methods ($n = 5$ replicates per group). The number of migrated siNeg cells was normalized to 1. Bar, 500 μm. **d** Supplementation of transsulfuration pathway metabolites partially prevents the resistance of HNF4α-depleted cells to MCR and sorafenib. Indicated cells were treated with or without 1 mM Ctt, 1 mM NaSH, or 10 mM taurine for 48 h in CM, then incubated in CM (left), MCR (middle), or CM containing 5 mM sorafenib (right) for additional 24 h. **e** Supplementation of metabolites in the transsulfuration pathway represses HNF4α depletion-induced cell migration. SiNeg and siHNF4α HepG2 cells were treated and analyzed for cell migration as described in Methods ($n = 5$ replicates per group). The number of migrated siNeg cells in CM was normalized to 1. Bar, 500 μm. **f** Addition of cystine partially rescues MCR-induced cell death in HepG2 cells but not in HNF4α-deficient cells. Indicated cells were cultured in CM, MCR, or MCR with 200 μM L-cystine (MCR + cystine) for 24 h ($n = 5$ replicates per group). For all bar graphs, values are expressed as mean ± s.e.m., two-tailed, unpaired Student's *t*-test, *$p < 0.05$.

Finally, genetic restoration of SAA metabolism also significantly reduced the mesenchymal characteristics of SNU449 cells. As shown in Fig. 8d and Supplementary 10d, overexpression of individual SAA enzymes, particularly BHMT, CBS, or CDO1, significantly suppressed cell migration in SNU449 cells similar to HNF4α overexpression. Again, at the transcriptional level, overexpression of individual SAA enzymes had distinct impacts on EMT changes (Fig. 8e). Consistent with observations in Fig. 8a, b, overexpressing CBS in SNU449 cells primarily suppressed the expression of a number of mesenchymal markers that were induced in HNF4α-deficient cells. Overexpression of BHMT not only repressed the expression of many mesenchymal markers but also induced the expression of two epithelial markers that were repressed in siHNF4α cells (Fig. 8e). Interestingly, overexpression of MAT1A or CDO1 induced expression of the majority of tested EMT markers despite their distinct activities on cell migration (Fig. 8e). In sum, these observations confirmed that SAA metabolism is an essential element in HNF4α-mediated stress sensitivity and EMT suppression.

## Discussion

Dietary methionine restriction has been shown to extend life span, reduce body fat, and improve insulin sensitivity through

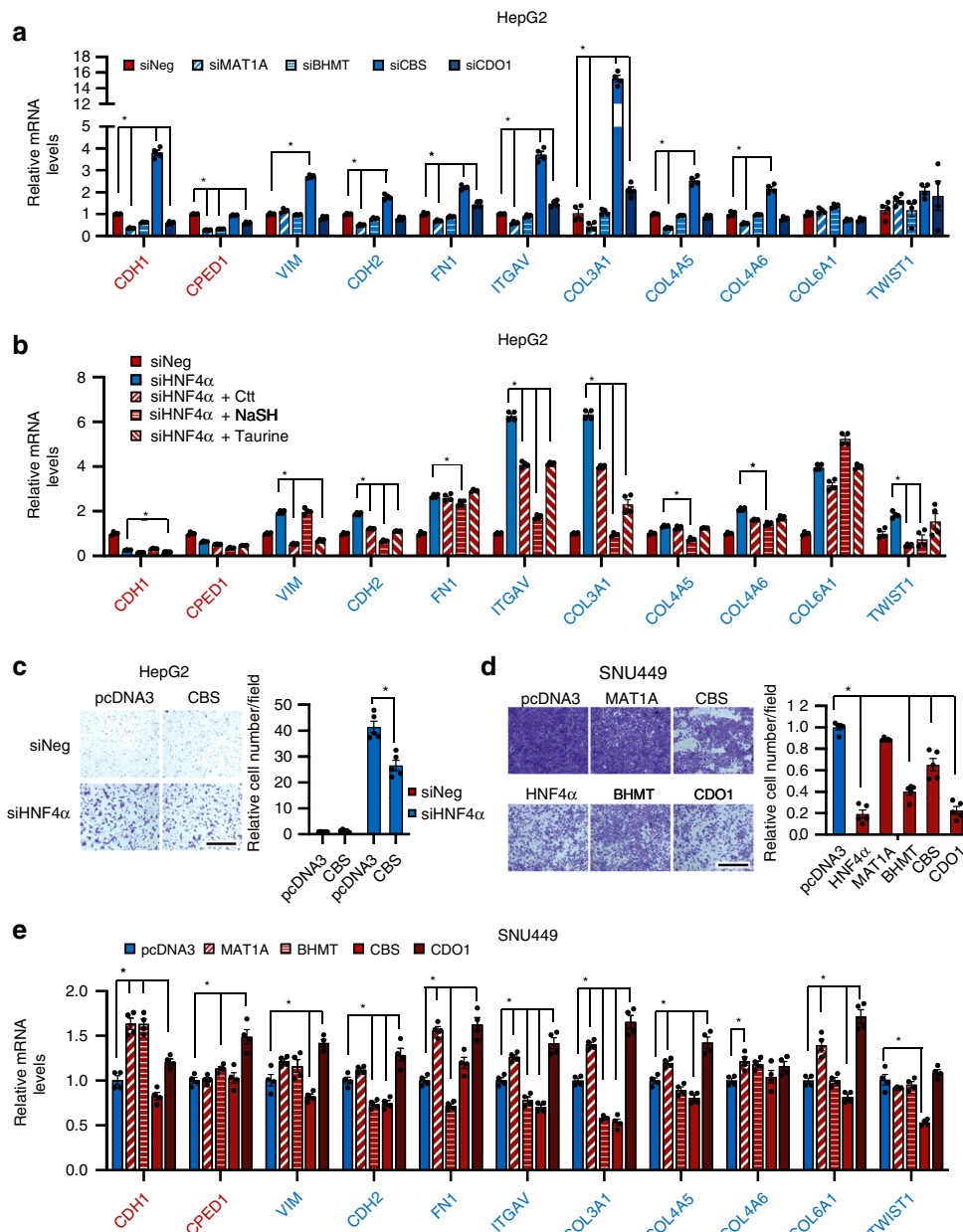

**Fig. 8 The transsulfuration pathway regulates EMT and migration of liver cancer cells. a** Knockdown of individual SAA enzymes in HepG2 cells has distinct impacts on expression of epithelial and mesenchymal markers. Indicated HepG2 cells were analyzed for expression of epithelial and mesenchymal markers by qPCR ($n = 4$ replicates per group). **b** Supplementation of metabolites in the transsulfuration pathway partially represses the expression of HNF4α depletion-induced mesenchymal markers in HepG2 cells. SiNeg or siHNF4α HepG2 cells were cultured in the complete medium with or without 1 mM Ctt, 1 mM NaSH, or 10 mM taurine and cultured for 48 h. The relative expression levels of the indicated epithelial and mesenchymal markers were analyzed by qPCR (n = 4 replicates per group). **c** Overexpression of CBS partially inhibits cell migration induced by HNF4α depletion. SiNeg and siHNF4α HepG2 cells were transfected with either an empty vector (pcDNA3) or a pcDNA3-CBS construct expressing CBS. Cells were then analyzed for cell migration in a transwell assay ($n = 5$ replicates per group). The number of migrated control cells (siNeg + pcDNA3) was normalized to 1. Bar, 500 μm. **d** Overexpression of SAA metabolic genes or HNF4α represses migration of HNF4α-negative SNU449 cells. SNU449 cells were transfected with either an empty vector (pcDNA3) or a pcDNA3 construct expressing the indicated SAA enzyme. Cells were then analyzed for cell migration in a transwell assay ($n = 5$ replicates per group). The number of migrated control cells (pcDNA3) was normalized to 1. Bar, 500 μm. **e** Overexpression of individual SAA enzyme in HNF4α-negative SNU449 cells has distinct impacts on expression of the expression of epithelial and mesenchymal markers. Control (pcDNA3) SNU449 cells, and SNU449 cells overexpressing individual SAA enzyme were analyzed for expression of epithelial and mesenchymal markers by qPCR ($n = 4$ replicates per group). For all bar graphs, values are expressed as mean ± s.e.m., two-tailed, unpaired Student's t-test, *$p < 0.05$.

various signaling pathways in animal studies[13,42,43,57,58]. Recent studies have also demonstrated methionine restriction as a powerful dietary intervention capable of inducing rapid and specific metabolic changes to influence cancer therapeutic outcomes[14]. However, we and others[15] have found that the response

of different human cancer cells to this nutritional manipulation is heterogenous (Fig. 3). In this study, we show that the heterogenous response of liver cancer cells to methionine restriction is due to, at least partially, their distinct HNF4α status. We provide evidence that the expression of key enzymes in SAA metabolism

is under transcriptional control by HNF4α. Consistently, HNF4α-negative mesenchymal liver cancer cell lines have rewired SAA metabolism and are more resistant to cell death induced by methionine/cystine restriction or sorafenib than HNF4α-positive liver cancer cells. Knocking down HNF4α in HNF4α-positive epithelial liver cancer lines impairs SAA metabolism, promoting epithelial-mesenchymal transition and increasing resistance to methionine restriction in vitro and in vivo (Fig. 6). We further show that overexpression of enzymes or supplementation of key metabolites in the transsulfuration pathway of SAA metabolism significantly restores the sensitivity of HNF4α-depleted liver cancer cells to methionine/cystine restriction or sorafenib treatment and inhibits cell migration (Fig. 7). Collectively, our findings not only identify a genetic master regulator of SAA metabolism, but also demonstrate that HNF4α-mediated transsulfuration is a key determinant of sensitivity to methionine/cystine restriction in liver cancer.

Development of liver cancer features mutations or aberrant expression of a number of cancer-associated genes, as well as hundreds of metabolic genes[59–61]. Our study identifies HNF4α-mediated SAA metabolism as a key mechanism of EMT suppression that contributes to nutritional and chemical sensitivity in liver cancer. We found that the expression levels of HNF4α and SAA metabolic genes are inversely correlated with those of the mesenchymal markers in HCC patients (Fig. 1b, d). Knockdown of HNF4α in epithelial liver cancer cells led to an impaired SAA metabolism that closely resembles mesenchymal liver cancer cells (Fig. 5). Moreover, knocking down HNF4α or SAA metabolic enzymes increased nutritional and chemical resistance (Fig. 6). Finally, deficiency of HNF4α or SAA enzymes promoted EMT characterized by reduced expression of epithelial markers and/or increased expression of mesenchymal markers (Figs. 6a and 8a, b), as well as enhanced cell migration (Fig. 7c). Importantly, restoring the expression of SAA metabolic genes suppressed EMT markers (Fig. 8e) and inhibited cell migration in mesenchymal liver cancer cells (Fig. 8d). Therefore, our study suggests that nutritional supplementation of SAA metabolites, such as Ctt, $H_2S$ precursors, and taurine, might offer therapeutic strategies to increase the sensitivity of liver tumors to dietary intervention (e.g., methionine restriction) or chemotherapy. It will be of great importance to test this possibility in future studies.

Our data further suggest that the transsulfuration pathway in SAA metabolism, particularly CBS-mediated production of Ctt and $H_2S$, may be one of the major effectors of HNF4α-mediated EMT suppression in liver cancer. CBS is tightly regulated by HNF4α transcriptionally (Fig. 4e, CBS Luc Reporter), and knocking down CBS in HepG2 cells most closely mimics HNF4α deficiency in promoting the resistance to methionine/cystine restriction and sorafenib (Fig. 7a), inducing mesenchymal morphology (Supplementary Fig. 7b), and reducing $H_2S$ production (Fig. 5c). Moreover, functional restoration of CBS activity in HNF4α-depleted liver cancer cells, either by supplementation of Ctt or $H_2S$ (Figs. 7e and 8b) or by overexpression of CBS (Fig. 8c–e), significantly alleviates stress resistance, cell migration, and EMT induced by HNF4α deficiency. However, despite the aforementioned observations, how CBS regulates tumor progression is still an open question. As one of the major enzymes producing $H_2S$ in mammalian cells, CBS catalyzes the initial and rate-limiting step in the transsulfuration pathway (Fig. 1a). Four metabolites involved in CBS-catalyzed reactions, Hcy, serine, Ctt, and $H_2S$, are bioactive metabolites involved in diverse biological functions including mitochondrial bioenergetics, redox homeostasis, DNA methylation, and protein modification[62]. Therefore, CBS could potentially affect a number of cellular processes, which in turn impact EMT and related drug resistance, and eventually cancer growth and progression. In liver cancer cells, a previous study has shown that CBS-generated $H_2S$ can induce autophagy and apoptosis through the PI3K/Akt/mTOR pathway[63]. Our data here suggest that both Ctt and $H_2S$ could be functional metabolites in CBS-mediated suppression of EMT (Figs. 7e and 8b), as CBS knockdown induces the expression of a number of mesenchymal markers, including *COL3A1* and *SNAI2* (Fig. 8a and Supplementary Fig. 10c). It will be interesting to further dissect the molecular mechanisms underlying these CBS-regulated alterations in future studies. It is also worth noting that the impact of CBS on tumor growth and progression is context-dependent, with CBS functioning as an oncogene in colon, ovarian and breast cancer but as a tumor suppressor in liver cancer[62,64]. Additional studies to investigate the interaction between HNF4α and CBS will therefore help to better understand the complex role of CBS in tumor biology.

Although our study uncovers a key role of HNF4α in regulation of SAA metabolism, it does not exclude the involvement of other factors in this process, particularly in human liver cancer cell lines. Human liver cancer lines are genetically highly heterogeneous. For example, considering HepG2, Huh7, and Hep3B, even though they are all epithelial liver cancer cells with high expression of HNF4α, HepG2 cells have normal expression of wild-type p53, Huh7 cells overexpress a mutant p53, while Hep3B cells are p53 null[65]. It will be interesting to find out whether various mutations in different liver cancer cells lines also contribute to distinct SAA metabolism features in future studies.

Our study has important translational implications as it identifies HNF4α as a potential biomarker for liver cancer patient selection in prospective clinical trials of dietary interventions with methionine restriction. We propose that patients with high HNF4α levels in tumors would be good candidates for such trials. Combination of methionine restriction with sorafenib treatment in these patients could produce promising survival benefits based on our in vitro results (Fig. 3e).

Our study has a technical limitation. Although we demonstrate that the MCR diet we used is able to suppress HNF4α-positive liver cancer growth in mice using two independent liver cancer models (xenograft model in Fig. 6h, i, and DEN/HFD model in Supplementary Fig. 8), we did not directly measure the methionine levels in blood and tumors in our setup and confirm that the MCR diet can indeed reduce circulating and tissue methionine. On the other hand, a 30-80% reduction of circulating methionine levels has been previously observed in both mice and humans after days to weeks of 80% dietary methionine restriction from independent research facilities[12,14,66]. Nevertheless, future analysis of circulating and tissue methionine concentrations during methionine-containing diet feeding, as well as their variance during normal diurnal cycle will help to better assess the therapeutic importance of dietary methionine restriction in the treatment of human liver cancer patients.

In summary, our study identifies a genetic regulator of hepatic SAA metabolism and a key determinant of sensitivity to methionine restriction and chemotherapy in liver cancer. Our findings may pave the way for therapeutic strategies against liver cancer and other HNF4α-associated human cancers.

## Methods

**TCGA database mining**. We downloaded the processed TCGA RNA-seq gene expression data for 50 normal and 373 tumor LIHC samples from The Cancer Genome Atlas–Data Portal (https://portal.gdc.cancer.gov/). We $\log_2$-transformed the normalized read counts (per million reads mapped) for RNA-seq data (all values less than 1 were assigned value 1 before transformation) but carried out no further normalization.

For correlation analysis, we computed the pair-wise Pearson correlation coefficient and the corresponding p-value between the expression levels of two genes using MATLAB. For data shown in Fig. 2b, we removed three outlier samples

in which HNF4A expression levels were more than 3 interquartile ranges (IQRs) below the first quartile among the 371 samples.

For survival analysis, TCGA data includes clinical information on characteristics of both patients (e.g., demographics, vital status at the time of report, treatment regimens, and clinical follow-up) and their tumors (e.g., disease-specific diagnostic/prognostic factors). The accurate stage of disease at the time of the TCGA biospecimen procurement was often not available. The pathological data such as primary tumor staging information were referenced to the patient's initial cancer diagnosis. Though ideally, we would like to measure survival time from initial diagnosis to death, we know that patients received their initial diagnoses before TCGA procured their biospecimens; thus, these patients have been at risk for a period before sample procurement. In addition, some patients may die after diagnosis but before their samples could be procured. Because of this lag between the initial cancer diagnosis and TCGA biospecimen procurement, our analysis is based on patients' survival from the time of TCGA biospecimen procurement to death or last follow-up. Specifically, the curated post-procurement survival is calculated as follows, post-procurement survival = days_to_last_contact – days_to_sample_procurement. If a patient has multiple follow-ups, we used the latest lost-to-follow-up date or the earliest death date. In addition, we filtered out one patient with negative post-procurement survival.

We calculated the coefficient estimate (beta value) and *p*-value using Cox proportional hazards regression implemented in MATLAB, and we visualized the survival distribution with Kaplan-Meier survival curves. Note that the Kaplan-Meier survival curves were generated using samples whose expression levels were among the top and bottom 33% of expression values for corresponding genes.

For clustering, we extracted the RNA-seq expression data of the 18 genes from the data above. For each gene, we standardized its expression levels across the 373 samples to a mean of 0 and standard deviation of 1 (z transformation). Next, any data points with a standardized value less than the negative of the maximum standardized value (i.e., 5.5) were assigned to −5.5. Only 14 data points out of 8206 data points were affected. This reassignment was to ensure that the colors in the heatmap were balanced. We then carried out a two-way hierarchical clustering analysis using the Euclidean distance metric for (dis)similarity measure and displayed the clustering results using a heatmap.

**Cluster analysis of liver cancer cell lines**. To analyze the association between HNF4α and SAA metabolic genes in liver cancer cells, we performed a hierarchical cluster analysis on the expression data of the 25 liver cancer cell lines downloaded from Broad Institute Cancer Cell Line Encyclopedia (CCLE) and of the 81 liver cancer cell lines downloaded from Liver Cancer Model Repository (LIMORE)[41] using MATLAB. We computed the Z-scores of the log$_2$(RPKM + 1) values across samples for the clustering analysis. For both sample clustering and gene clustering, we applied *spearman* distance as similarity measure between every pair of objects in the dataset, then we generated binary hierarchical cluster tree using *linkage* function which employs *average* method for computing the distance between clusters. Subsequently, the clusters were formed using *cluster* function. To assess the hierarchical cluster tree, we computed the cophenetic correlation coefficient which measures how accurate the clusters reflect the data. The cophenetic correlation coefficient is 0.86 in the CCLE dataset and 0.89 in the LIMORE dataset (with 1 as the optimal value), indicating reasonably well-clustered result.

**Cell culture**. Five liver cancer cell lines, Huh7, Hep3B, HepG2, and SNU449, were obtained from the Cell Repository at the Tissue Culture Facility of the UNC Lineberger Comprehensive Cancer Center, all of them were originated from ATCC. SNU475 was purchased directly from ATCC. Huh7, Hep3B, and HepG2 were maintained in DMEM (ThermoFisher Scientific) supplemented with 10% FBS (Hyclone). SNU449 and SNU475 were maintained in RPMI-1640 (ThermoFisher Scientific) supplemented with 10% heat inactivated FBS. All liver cancer cell lines were cultured in DMEM (GIBCO) supplemented with 10% FBS (Hyclone) for experiments.

Normal human primary hepatocytes (ThermoFisher Scientific) were thawed in the Cryopreserved Hepatocyte Recovery Medium, centrifuged, then resuspended and plated in the plating medium (Williams' Medium E without phenol red supplemented with Hepatocyte Plating Supplement Pack). After incubation at 37 °C for 6 h, the medium was replaced with incubation medium (Williams' Medium E without phenol red supplemented with Hepatocyte Maintenance Supplement Pack). After overnight incubation, the cells were used for the experiments. All the cell culture reagents for the Normal human primary hepatocyte were purchased from ThermoFisher Scientific.

To knock down HNF4α and SAA genes, siRNAs against HNF4α (SiHNF4α#1, s533433; SiHNF4α#2, s6698), individual SAA gene (siMAT1A, s8523; BHMT, s1981; CBS, s528455; CTH, s3711; or CDO1, s2856), and a negative siRNA (siNeg, 4390843) were purchased from ThermoFisher Scientific and transfected into liver cancer cells and normal human hepatocytes with Lipofectamine RNAiMAX (ThermoFisher Scientific).

To overexpress HNF4α and SAA genes, pcDNA3 vector (ThermoFisher Scientific) and pcDNA3 cDNA ORF clones of HNF4α (OHu25290D), MAT1A (OHu18604D), BHMT (OHu17861), CBS (OHu26151), and CDO1 (OHu17128) from Genscript USA were transiently transfected into the indicated cells with Lipofectamine 3000 (ThermoFisher Scientific).

**Metabolomics analysis**. To quantitatively analyze metabolic profiles in siNeg HepG2, siHNF4α HepG2, and siNeg SNU449 cells, cells were cultured in regular DMEM + 10% FBS medium. Metabolites were then extracted and analyzed as described previously[67]. Briefly, cells cultured in 6-well plates (triplicates) were extracted with 1 ml ice-cold extraction solvent (80% methanol/water) by incubation at −80 °C for 10 min and centrifugation at 20,000 × *g* for 10 min at 4 °C. The supernatant was transferred to a new Eppendorf tube and dried in vacuum concentrator. The dry pellets were stored at −80 °C for liquid chromatography with high-resolution mass spectrometry analysis. Samples were reconstituted into 30–60 μl sample solvent (water:methanol:acetonitrile, 2:1:1, v/v/v) and were centrifuged at 20,000 × *g* at 4 °C for 3 min. The supernatant was transferred to liquid chromatography vials. The injection volume was 3 μl for hydrophilic interaction liquid chromatography (HILIC).

High-performance liquid chromatography was performed essentially as described previously[67]. Specifically, an Ultimate 3000 UHPLC (Dionex) was coupled to the Q Exactive-Mass plus spectrometer (QE-MS, Thermo Scientific) for metabolite separation and detection. For additional polar metabolite analysis, a HILIC method was used, with an Xbridge amide column (100 × 2.1 mm internal diameter, 3.5 μm; Waters), for compound separation at room temperature.

Mass spectrometry and data analysis: the QE-MS was equipped with a HESI probe, and the relevant parameters were: heater temperature, 120 °C; sheath gas, 30; auxiliary gas, 1; sweep gas, 3; spray voltage, 3.6 kV for positive mode and 2.5 kV for negative mode. Capillary temperature was set at 320 °C, and the S-lens was 55. A full scan range was set at 60 to 900 (*m/z*) when coupled with the HILIC method. The resolution was set at 70,000 (at *m/z* 200). The maximum injection time was 200 ms. Automated gain control was targeted at 3 × 10$^6$ ions. Liquid chromatography–mass spectrometry peak extraction and integration were analyzed with commercially available software Sieve 2.0 (Thermo Scientific). The integrated peak intensity was used for further data analysis.

Metabolite pathway analysis was performed in the Pathway Analysis module of MetaboAnalyst 4.0[68]. All metabolites were normalized with total analyzed cell numbers for analysis. For metabolite clustering analysis, average linkage hierarchical clustering was performed in the Statistical Analysis module of MetaboAnalyst 4.0 using Euclidian distance as a similarity metric. For PCA analysis, all metabolites were used following quantile-normalization in the Statistical Analysis module of MetaboAnalyst 4.0.

**Targeted analysis of metabolites from the SAA metabolic pathways**. To confirm the abundance of SAA metabolites in siNeg HepG2, siHNF4α HepG2, and siNeg SNU449 cells, metabolites were extracted as in above Metabolomics analysis. Mass spectrometry data were acquired on a Q Exactive Plus mass spectrometer (QE-MS, ThermoFisher Scientific) and Vanquish (ThermoFisher Scientific) UHPLC system. Reverse-phase chromatography was performed using a CORTECS C18 guard column (5 mm × 2.2 mm i.d., 1.6 mm; Waters) and CORTECS C18 analytical column (100 × 2.1 mm i.d., 1.6 mm; Waters) with solvent A being 5 mM ammonium formate in water (pH 6.5) and solvent B being methanol. The LC gradient included a 0.5-min hold at 0%B followed by a ramp from 0 to 42% B over the next 6 min followed by a ramp to 95% over the next minute. A 3-min hold at 95% was followed by a return to 0% B over the next 0.5 min. The run was completed with a 5-min recondition at 0% B. For the mass spectrometry, a PRM method was employed with an included list for the masses of the metabolites of interest and their optimized normalized collision energies (Supplementary Table 4). The QE-MS was equipped with a HESI source used in the positive ion mode with the following instrument parameters: sheath gas, 40; auxiliary gas, 10; sweep gas, 1; spray voltage, 3.5 kV; capillary temperature, 325 °C; S-lens, 50; scan range (*m/z*) of 70 to 1000; MS resolution 70,000; 2 *m/z* isolation window; MSMS resolution: 17,500; MS automated gain control (AGC), 3 × 10e6 ions; MSMS AGC, 2 × 10e5 ions; and a maximum injection time of 200 ms. Mass calibration was performed before data acquisition using the LTQ Velos Positive Ion Calibration mixture (Pierce). PRM data were processed using the Qual Browser application in the Xcalibur software suite (Thermo Scientific). Extracted ion chromatograms for fragment ions were drawn for each compound in their respective channels and areas under the peak calculated and used to represent the relative abundance of the metabolites in the samples.

**Cell survival and caspase assays**. WST-1 (Sigma) or Caspase-Glo 3/7 assay (Promega) were used to measure proliferation and apoptosis of liver cancer cells, respectively. WST-1 and caspase assays were performed according to manufacturer's instructions.

**ROS and cell cycle assays**. Oxidative stress was measured with CellROX Green Flow Cytometry Assay Kit (ThermoFisher Scientific) and Cell proliferation was measured with Click-iT EdU Flow Cytometry Assay Kit (ThermoFisher Scientific) according to the manufacturers' instructions.

**Lead sulfide assay for H2S production**. H$_2$S production of HepG2 and SNU449 cells was detected by the lead sulfide method described in Hines et al.[69]. Briefly, cells were cultured in 96 well plates in growth media supplemented with 10 mM L-cysteine and 10 μM pyridoxal 5-phosphate hydrate (Sigma). A piece of 703 style

Whatman filter paper (VWR), soaked in 20 mM lead acetate (Sigma) and dried, was placed over the culture wells and covered with the plate lid with a heavy object on the top. The cells were cultured in a $CO_2$ incubator at 37 °C for 24 h. The formation of lead sulfide indicated by the dark circles on the filter paper was recorded with a digital camera.

**Chromatin Immunoprecipitation**. Chromatin immunoprecipitation (ChIP) analysis was performed as previously described by Shimbo et al.[70] with some modifications. The anti-HNF4α antibodies were obtained from Abcam (ab41898, 2 μg antibody per 25 μg of genomic DNA). Sequences of ChIP-PCR primers can be found in Supplementary Table 5.

**Immunoblotting**. Protein samples were separated by SDS-PAGE, transferred to PVDF membrane, and probed with following antibodies: HNF4α (Santa Cruz, sc-374229, clone H-1, Lot B2417, 1:500), BHMT (Santa Cruz, sc-390299, clone H-7, Lot B1913, 1:500), CBS (Santa Cruz, sc-133154, clone B-4, Lot D0918, 1:1000), CDO1 (ThermoFisher Scientific, PA5-38005, Lot UD2757664A, 1:1000), MAT1A (Abcam, ab129176, Lot GR91375-9, 1:1000), CTH (Proteintech Group, 12217-1-AP, Lot 00076987, 1:1000), Puromycin (Millipore Sigma, MABE343, clone 12D10, Lot 3379285, 1:1000), Actin (Millipore Sigma, MAB1501, clone C4, Lot 3132961, 1:10,000), HRP-conjugated alpha Tubulin (Proteintech, HPR-66031, mouse monoclonal, Lot 21000018, 1:2000).

**Luciferase assay**. To directly analyze the transcriptional regulation of SAA enzyme expression by HNF4α, firefly luciferase reporters driven by the human *MAT1A* (NG_008083, 1684-1890), *BHMT* (NG_029156, 2128-2282), and *CBS* (NG_008938, 12674-12911) promoters containing either wild type (WT) or mutant HNF4α binding site were cloned into pGL3 basic vector. The mutants were constructed with QuikChange II XL Site-Directed Mutagenesis Kit (Agilent Technologies). For *MAT1A*, HNF4α binding site gcttcagagttca was mutated to gcttttttttga; For *BHMT*, HNF4α binding site agatcagagaaca was mutated to gggggggggggggg; For *CBS*, HNF4α binding site ccgggtgagggtcaaag was mutated to gggggggggggggaag. The WT or mutant plasmids were then transfected into HNF4α-negative SNU449 cells together with the control pRL-TK (Renilla Luciferase, Promega), and the control pcDNA3 vector or pcDNA3-HNF4α construct. Cells were cultured for 24 h and the luciferase activity was measured using the Dual-Luciferase Reporter Assay System (Promega). The final firefly luciferase activity was normalized to the co-expressed renilla luciferase activity.

**Animal experiments**. All experimental mice were housed in rooms with a constant temperature (19–23 °C) and humidity (55% ±10%) and a 12-h light/dark cycle, and had free access to water and food.

To test whether HNF4α confers sensitivity of liver tumors to methionine restriction in vivo, $5 \times 10^6$ siNeg and siHNF4α HepG2 cells were injected subcutaneously into each flank of female NU/J mice (#002019, Jackson Laboratory). Mice were monitored twice weekly for tumor growth and health status. Tumor length and width were measured by caliper and tumor volume was calculated using the formula $V = \text{length} \times \text{width}^2/2$. When the average tumor volume reached 200 mm³ (in about 2–3 weeks) mice were randomized to two treatment groups, one group was fed with a methionine deficient diet containing 0.172% DL-methionine (cystine-free) and the other group was fed with a control diet containing 0.86% DL-methionine (cystine-free) ad libitum with free access to water. Methionine deficient diet (#510029, MCR) and the control diet (#510027, CTR) were purchased from Dyets, Inc. (Bethlehem, PA). Mice were sacrificed when their total tumor volume reached 2000 mm³ according to the approved animal protocol (in 3 weeks), so the total experimental timeframe was around 5 weeks. This animal work was approved by the Institutional Animal Care and Use Committee of the National Institute of Environmental Health Sciences.

To examine the impact of MCR on liver tumorigenesis in vivo, 2-week male C57BL/6 mice were injected intraperitoneally with diethylnitrosamine (DEN) at 25 mg/kg. From 4-week old, mice were fed with a high-fat diet (D12492, Research Diets, USA) for 5 months, switched to above MCR or CTR diets for 4 months, and then switched back to HFD for additional 2 months. The livers were removed and separated into different lobes. Visible tumors were counted and measured. Large lobes were used for immunostaining and remaining lobes were separated into tumor and non-tumor tissues and stored at −80 °C for further analysis. This animal work was approved by the Animal Care and Use Committee of the Shanghai Jiao Tong University School of Medicine.

**Immunohistochemistry**. Immunohistochemical staining of formalin-fixed, paraffin-embedded xenografted tumor tissues was performed using rabbit monoclonal Ki-67 antibody (Abcam, Cambridge, MA, Cat# ab16667, Lot# GR3185488-3, 1 mg/ml) at a 1:150 dilution. Further, the sections were incubated with Rabbit-on-Rodent HRP-Polymer (Biocare Medical, Concord, CA) for 30 min at room temperature. The antigen-antibody complex was visualized using 3-diaminobenzidine (DAB) chromogen (Dako North America, Inc., Carpinteria, CA) for 6 min at room temperature. Finally, the sections were counterstained with hematoxylin, dehydrated through graded ethanol, cleared in xylene, and coverslipped. The ratio of Ki67-positive tumor area was calculated using the ImageJ 2.0.0 (Fiji) software.

**Quantitative real-time PCR**. Total RNA was purified from cells or tissues using RNeasy Mini kit (Qiagen) or Trizol reagent (Ambion). cDNA synthesized with the High-Capacity cDNA Reverse Transcription Kit (ThermoFisher Scientific) was subjected to quantitative real time PCR with iQ SYBR Green Supermix (Biorad). All the data were normalized to 18 s RNA. Sequences of qPCR primers can be found in Supplementary Table 5.

**Cell migration assay**. To analyze the cell migration ability, HepG2 cells were transfected with siRNAs and SNU449 cells were transfected with overexpression plasmids for 48 h. Cells were then trypsinized and plated with serum-free DMEM on the top of the membrane in a transwell insert with 8 μM pore size (Corning). The inserts were placed in the wells of a 24-well plate filled with 10% FBS-containing DMEM, cultured for 24 h and then stained with crystal violet (Sigma). A cotton-tipped applicator was used to remove the non-migrated cells on the top side of the insert. Migrated cells were imaged with a microscope and quantified with the ImageJ 2.0.0 (Fiji) software.

To analyze the impact of metabolites in the transsulfuration pathway on HNF4α depletion-induced cell migration, HepG2 cells were transfected with control siRNAs (siNeg) or siRNAs targeting HNF4α (siHNF4α) in the complete medium. On the next day, cells were switched to the complete medium with or without 1 mM Ctt, 1 mM NaSH, or 10 mM taurine and cultured for 48 h. Cell migration was then analyzed in a transwell assay for 24 h.

**Quantification statistical analysis**. Values are expressed as mean ± standard error of the mean (s.e.m.) from at least three independent experiments or biological replicates, unless otherwise indicated in the figure legend. Significant differences between the means were analyzed by the two-tailed, unpaired, non-parametric Mann–Whitney test for in vivo experiments and the two-tailed, unpaired Student's *t*-test for in vitro experiments, and differences were considered significant at $p < 0.05$. Data were analyzed using Prism Software 8.0 (GraphPad) or Microsoft Office Excel (Version 16.16.23).

**Reporting summary**. Further information on research design is available in the Nature Research Reporting Summary linked to this article.

## Data availability

The TCGA RNA-seq gene expression data were downloaded from The Cancer Genome Atlas–Data Portal http://gdac.broadinstitute.org/runs/stddata__2016_01_28/data/LIHC/20160128/). The RNA-seq data of different human liver cancer cell lines were downloaded from Broad Institute Cancer Cell Line Encyclopedia (CCLE, https://portals.broadinstitute.org/ccle/data; file name: CCLE_RNAseq_rsem_genes_tpm_20180929.txt.gz) and from Liver Cancer Model Repository (LIMORE, https://www.picb.ac.cn/limore/batch; download link: Gene expression profiles of 81 cell lines). Metabolomics data are provided in Supplementary Tables. Source data are provided with this paper.

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

## Acknowledgements

We thank Drs. Paul Wade and Stephen Shears and members of the Li laboratory for critical reading of the manuscript. We also thank the Comparative Medicine Branch of National Institute of Environmental Health Sciences for support with animal experiments and Drs. Masahiko Negishi and Muluneh Fashe for providing normal human hepatocytes. This research was supported by the Intramural Research Program of National Institute of Environmental Health Sciences of the NIH to X.L. (Z01 ES102205).

## Author contributions

Q.X. designed the study, performed experiments, analyzed data, and wrote the manuscript. Y.L., K.K., and L.L. performed bioinformatic analysis of TCGA, CCLE, and LIMORE databases. X.G., J.L., and J.W.L. performed global metabolomic analysis and determined medium methionine and cystine concentrations by LC-MS. J.G.W. and L.J.D. performed targeted analysis of SAA metabolites by LC-MS. L.T. and X.T. analyzed the impact of MCR diet on liver tumorigenesis in the DEN/HFD model. M.J. assisted the mouse xenograft experiment. I.S. guided and designed the study, analyzed data, and wrote the manuscript; X.L. guided, designed, and coordinated the study, analyzed data, and wrote the manuscript. All authors critically reviewed the manuscript.

## Competing interests

The authors declare no competing interests.
