## [Peer Review File · Nature Communications]

Reviewers' comments:

Reviewer #1 (Remarks to the Author); expert on diet restriction and metabolism, mouse models:

The study describes an analysis of sulfur amino acid metabolism in hepatocellular carcinoma. This is a very interesting area and the suggestion that interventions can be identified to increase the sensitivity of tumors to dietary limitation of methionine is very interesting. In general, the data that are presented are compelling and the correlations are impressive. This is potentially a very important study.

The first three figures show correlations (albeit quite convincing) between HNF4a, expression of SAA metabolism enzymes, EMT and sensitivity to methionine/cysteine starvation. Most of the subsequent mechanistic data is limited to the analysis of one cell line treated with siRNA. Expanding and strengthening these results would greatly improve the study – I have some suggestions below.

Secondly, the study fails to provide any insight into the basis for the resistance of the HNF4a non-expressing cells to methionine and cysteine starvation. It would seem the cells don't grow under these conditions (Figure 3C) but why don't they die? The title states this is due to regulation of SAA metabolism, but it's not clear to me how impairing SAA metabolism would promote cell survival (or drives EMT). The authors do make some suggestions in the discussion and I think some insight into this would greatly enhance the study.

Specific points

1. In Figure 4, knock down of HNF4a in HepG2 cells is shown to decrease the expression of the SAA metabolism genes. To support the data in Figure 3 it would be useful to show the expression of these genes in all the HCC cell lines including HNF4a depleted Huh7 and Hep3B cells (which are described in Figure 6E).
2. The authors show nicely the effect of HNF4a re-expression on reporters in SNU448 cells (Figure 4D) but I think it would be important to show that there is an increase in expression of the endogenous SAA genes in these cells too.
3. The analysis of siRNA KD in HepG2 cells is fine, but the authors should also validate the key responses in the other HNF4a expressing cells. As siRNAs can have off target effects (and the authors seem only to use a mixture of 2 siRNAs with no rescue to validate the specificity of the response), a different approach to knock down, such as CRISPR, would also help to strengthen the results.
4. What happens to the metabolite profiles (as shown in Figure 5) in the SNU448 cells re-expressing HNF4a?
5. The use of siRNA treated cells in a 25-day tumorigenesis assay seems difficult – despite the apparent maintenance of reduced mRNA levels in the tumors (Figure S4B). It would appear from Figure 6F that depletion of HNF4a does not impact the survival of HepG2 or Hep3B cells in fed conditions, so it would be possible to generate stable HNF4a depleted lines. I think this needs to be done – at least for the in vivo studies – and would help to validate the in vitro work (see above).
6. Are SNU449 and SNU475 tumors insensitive to methionine restriction?
7. In Figure 7 the authors focus on CBS as a key mediator of the effect of HNF4a on EMT. But what is the effect of CBS depletion on SAA metabolism and the survival of cells under methionine and cysteine depletion? This would seem to be a critical point for the study.

Reviewer #2 (Remarks to the Author); expert on methionine/sulfur metabolism:

The manuscript by Xu et al. investigates a potential liability in some hepatocellular carcinoma (HCC) cells involving sulfur metabolism. It is known that decreased expression of sulfur amino acid

(SAA) pathways correlates with increased malignancy in HCC. Based on an analysis of RNAseq data from 373 HCC tumors and 50 normal samples in TCGA, the authors found a positive correlation between HNF-4a expression and expression of SAA metabolism pathways, and a negative correlation between expression of HNF-4a and markers of a mesenchymal phenotype, such as TWIST1. RNAseq data from 25 different established HCC cell lines showed similar correlations. Subsequent investigations were performed on established HCC cell lines, in particular the HNF4a-high expressing cell lines HepG2, HUH7, and Hep3B217, and the HNF4a-low HCC lines SNU449 and SNU475. The authors found that the HNF4a-high lines, in addition to having higher expression of SAA metabolism genes and lower expression of mesenchymal marker genes, had higher sensitivity to Met/Cys restriction, higher basal H₂S production, more of an epithelial phenotype, and higher sensitivity to Sorafenib. ChIP analyses indicate HNF4a interacts directly with 5 genes encoding SAA metabolic functions. Knockdown of HNF4a in HepG2 induced the cells to take on an SNU449-like phenotype, including diminished expression of SAA metabolism genes, diminished expression of epithelial markers, diminished sensitivity to Met/Cys restriction, diminished sensitivity to Sorafenib, increased expression of mesenchymal markers, and increased cell migration. In a mouse xenograft model, HepG2 cells expressing siRNA for HNF4a were less impacted by a Met-restricted diet than were tumors from HepG2 cells expressing a control siRNA. The paper identifies HNF4a as a regulator of SAA metabolism and perhaps also of a mesenchymal phenotype, and the authors conclude HNF4a, via regulating SAA metabolism, determines sensitivity of HCC to Met restriction.

Strengths of the paper.

This is, to my knowledge, the first evidence of HNF4a playing a direct role in regulating the transcription of the reported SAA metabolism genes. The complementary analyses of expression profiles, ChIP data, and knockdown data are compelling. In particular, the correlations between HNF4a and MAT1A, BHMT, and perhaps others in the large TCGA dataset matches pretty well with the ChIP and knockdown data. The robustness of the direct role of HNF4a in this regulation, however, breaks down somewhat in the established cell lines. For example, in comparing the correlation of HNF4a expression and some of its established targets (HNF1A, Alb) versus to the SAA targets identified in this report (MAT1A, BHMT, CBS - Fig. 1E), there are quite a few unexplained outliers, wherein HNF4a expression patterns match well with characteristics of the established targets, but not the putative SAA targets, in certain cell lines. The presence of outliers, both in the cultures and the TCGA data might be as important as the overall statistical correlations in making generalizations about HCCs.

Weaknesses of the paper.

Major:

1) Living cells cannot live without a usable source of exogenous sulfur. The cell culture experiments here are reportedly done under conditions of 0 mM Met and 0 mM cystine, which are the only two normal nutritional sulfur amino acids, and are the only two sulfur-containing molecules, in standard DMEM. In Met/Cys-free DMEM, there are no sulfur-containing molecules. Experiments go for up to 24 h under these conditions. Without Met and Cys, not only can there be no protein synthesis, but also no synthesis of any sulfur-containing molecules, including SAM, GSH, CoA, FeS-clusters, and others. Clearly there must have been contaminating sources of sulfur-containing nutrients in the experiments presented, such that the actual conditions were > 0 mM Met/Cys. Possible contaminating sources could have been from small molecules in the serum (Methods do not specify that serum was dialyzed), components of other media supplements, proteolysis of serum proteins, or proteolysis of self via autophagy or similar processes. The actual sources and amounts of nutritional sulfur in these experiments needs to be quantified and presented.

2) Met is essential (not mentioned in the paper). Regardless of cystine availability, human cells cannot survive without an exogenous source of Met. Again, the conditions in this study cannot truly be 0 mM Met/Cys, as reported. Interestingly, Thr is also essential (also not mentioned), and Fig. 3D shows the expected lethality under Thr-restriction. This even further confounds

comprehension of what source of Met is rescuing the cells in this study, as proteolysis or autophagy would each yield Met and Thr.

3) Elimination of the Cys source along with Met from the medium was not justified, since having a Cys source cannot provide Met. Would similar results be obtained with only Met restricted?

4) All HNF4a-high cell lines were grown in DMEM + 10% FCS; all HNF4a-low cell lines were grown in RPMI + 10% heat-inactivated FCS. Experiments were done in amino acid-defined DMEM + 10% FCS. Therefore, in the experiments, the HNF4a-low, but not the -high, cell lines were exposed to media conditions different from what they are adapted to.

5) A mechanistic connection between the SAA metabolism pathways and sorafenib is not discussed in the paper, making it difficult to follow the logic behind why a kinase inhibitor should interact with Met restriction.

6) The conclusions of HNF4a-high and -low cell lines having different susceptibilities to sorafenib under Met/Cys restriction is not supported by data. Fig. 3E shows that the biggest difference in survival occurs at 0 μ M sorafenib. As doses of sorafenib increase, differences between Met/Cys restricted and Met restored conditions diminish for all cell lines.

7) The impacts of Met/Cys restriction on cellular levels of Met, Cys, or GSH, and on protein synthesis rates, proliferation, or oxidative stress, are not shown in the cell culture models.

8) The authors do not show that Met-restriction in the xenograft models affects the amount of available Met in circulation in the animals. The condition used (chow containing 0.172% Met) is only 2- to 4-fold below Met levels in many standard rodent chow formulations. Additional Met might come from gut microbes. The mouse studies should be supported by measurements of Met in circulation and in tissues.

Minor weaknesses

1) It is well established that phenotypically more differentiated HCCs are less malignant and less metastatic. HNF4a is a major determinant of the differentiated state of hepatocytes and of hepatocellular carcinomas. Other physical and molecular characteristics of differentiated hepatocytes include (1) strong expression of hepatocyte-specific genes (e.g.s Alb and HNF1A) and many SAA metabolism genes (MAT1A, BHMT, CBS, CTH); and (2) an epithelial morphology. Although most correlations between SAA metabolism, malignancy, metastasis, and HNF4a expression reported here are neither new nor surprising, it is novel and satisfying that the presented study suggests a direct regulatory connection between HNF4a and both the upregulation of SAA metabolism genes and the epithelial phenotype. It is, however, an unnecessary confusion for the field for this study to re-name "differentiated" and "undifferentiated" HCCs as "epithelial" and "mesenchymal", respectively.

2) Cysteine and cystine are confused at several places in the text.

3) "Relative enrichment" (or similar) on the Y-axis of several figures is not defined. Since the scales start at 0, it suggests this is a scale for which a zero-value is possible (i.e., not a ratio or fold-change).

4) Figs 2A and 5G are unclear. "Impact" is not defined, nor are there descriptions of what is signified by dot-color, -intensity, or -size.

5) ATCC describes HepG2 cells as "non-tumorigenic in immune compromised mice"; yet xenograft experiments presented in Fig. 6 contradict this. High quality histology of the tumors is not shown. Markers do not validate the origins of the tissue shown. Some further explanation to resolve this

discrepancy would be helpful.

Reviewer #3 (Remarks to the Author); expert on HCC and transcription:

Comments for the authors

The authors aimed to identify importance of HNF4A-SAA metabolism axis involved in methionine restriction in HCC by bioinformatics analyses, metabolomics, and molecular and cellular characterizations. Expression of HNF4A was reduced in HCC patients in TCGA dataset, positive correlation between HNF4A and SAA enzymes was also found, and low expression of key SAA enzymes showed poor prognosis in HCC patients. Similar results were observed in HCC or hepatoblastoma cell lines, and HNF4A was found to directly regulates expression of SAA enzymes. Suppressed expression of HNF4A or SAA enzymes in HNF4A-positive HepG2 cells showed increased resistance to methionine restriction and sorafenib, and induced cell migration and expression of mesenchymal markers. These data indicate that HNF4A-SAA metabolism axis play an important role to determine the sensitivity of HCC to methionine restriction. This is very interesting study, but the authors should explain, discuss, and/or demonstrate the following points to strength the manuscript.

Major

- (1) Figs. 1B and C; positive correlation between HNF4A and SAA enzymes is also found in both viral-HCC and non-viral HCC?
- (2) Fig. 1D; negative correlation between SAA enzymes and TWIST1 was found, but the authors do not use TWIST1 as an EMT marker in further Figures (Figs. 6A, 7E, and S5C). Why?
- (3) The authors mainly use HepG2 cells as HCC cells. However, HepG2 cells are hepatoblastoma cells, but not HCC cells (Lopez-Terrada, D et al, Human Pathol, 40, 1512-1515, 2009). In addition, protein expression of CBS is almost same between HepG2 and SNU449 (Fig. 1G). Protein expression of HNF4A and most SAA enzymes is highest in Huh7 and lowest in SNU475 among five cell lines. The authors should mainly use Huh7 and SNU475 as epithelial and mesenchymal cells, respectively. Otherwise, the authors should use both HepG2 and Huh7 as epithelial cells, and both SNU475 and SNU449 as mesenchymal cells to prove generality.
- (4) Fig. 3E; why did Met/Cys-depleted HepG2 cells show resistance to cell death induced by sorafenib compared to Huh7 and Hep3B cells? Because of hepatoblastoma cells?
- (5) Fig. 4A; the authors should show ChIP-IgG of SNU449.
- (6) Fig. 4D; luc activity of wild-type MAT1A promoter was induced about 2.5-fold by HNF4A, but the mutated promoter was also induced about 2.5-fold by HNF4A, meaning that the mutated HNF4A bind site is not functional. Are there any functional HNF4A binding sites in the MAT1A promoter as shown in Fig. S3? Also, expression of MAT1A, BHMT, and CBS mRNA and/or protein is induced by HNF4A in SNU449 cells?
- (7) Fig. 6A; the authors should investigate expression of epithelial and mesenchymal markers, cellular morphology, cell migration, and apoptosis in HNF4A-overexpressed mesenchymal cells.
- (8) Fig. 6G; expression of HNF4A in Hep3B is much lower compared to HepG2 and Huh7. Why did the authors use Hep3B, but not Huh7? In Fig. 1G, expression of HNF4A in Huh7 is higher than that in HepG2, but why is expression of HNF4A in Huh7 lower than that in HepG2 (Fig. 6E)?
- (9) Fig. 6H and I; there was significant difference in Ki67+ area between siHNF4A MR and siNeg MR groups (Fig. 6I), but were there significant difference in tumor volume between siHNF4A MR and siNegMR between these groups (Fig. 6H)?
- (10) Fig. 7B; the authors mentioned that knockdown of CBS and CDO1 increased stress resistance and induced cell migration. Did knockdown of each SAA enzyme enhance mesenchymal cellular morphology as shown in Fig. 6B?
- (11) Fig. 7E; why did knockdown of CBS elevate expression of CDH1 in spite of elevated

expression of several mesenchymal markers? The authors should show whether enhanced mesenchymal cellular morphology does occur in CBS-knockdowned HepG2 cells.

(12) Overexpression of individual SAA enzyme suppressed EMT in SNU449?

(13) The authors should show whether individual SAA enzyme is a direct target of HNF4A using human normal primary hepatocytes.

(14) It is not clear whether SAA enzymes such as CBS is involved in HCC progression because this research is mainly based on data using epithelial and mesenchymal cell lines. Thus, it is important to investigate whether deficiency of HNF4A-SAA metabolism axis is critical for HCC progression using HCC model experiments using wild-type mice, liver-specific Hnf4a-null mice, and/or SAA enzyme (especially CBS)-deficient mice.

Minor

(1) Fig. S1A; expression of BHMT2 is significantly suppressed in HCC compared to normal liver, but there's no description of BHMT2 hereafter. BHMT2 is not important for SAA metabolism in liver compared to BHMT?

(2) Figs. S1B and S2B legends; there's no explanation of blue and red bars.

(3) Fig. 3A; there's no explanation of blue bar.

(4) Line 189; "...displayed elevated basal levels of many of these genes..." should be modified as follows. "...displayed elevated basal levels of ATF4 and CHOP..."

(5) Fig. S3A-C legend; the authors should provide a detailed description.

(6) Line 265; "49 metabolites" should be corrected to "48 metabolites"?

Reviewer #1 (Remarks to the Author); expert on diet restriction and metabolism, mouse models:

The study describes an analysis of sulfur amino acid metabolism in hepatocellular carcinoma. This is a very interesting area and the suggestion that interventions can be identified to increase the sensitivity of tumors to dietary limitation of methionine is very interesting. In general, the data that are presented are compelling and the correlations are impressive. This is potentially a very important study.

We would like to thank the reviewer for the positive comments on the importance of this study.

The first three figures show correlations (albeit quite convincing) between HNF4a, expression of SAA metabolism enzymes, EMT and sensitivity to methionine/cysteine starvation. Most of the subsequent mechanistic data is limited to the analysis of one cell line treated with siRNA. Expanding and strengthening these results would greatly improve the study – I have some suggestions below.

We appreciate these constructive comments. We have performed additional experiments in additional cell lines to expand a number of analyses, including the impact of HNF4 α knockdown on the expression of SAA genes (new Supplementary Fig. 4d and Fig. 4d) and EMT markers (new Supplementary Fig. 6a) in Huh7, HepB3, and normal human hepatocytes.

Secondly, the study fails to provide any insight into the basis for the resistance of the HNF4a non-expressing cells to methionine and cysteine starvation. It would seem the cells don't grow under these conditions (Figure 3C) but why don't they die? The title states this is due to regulation of SAA metabolism, but it's not clear to me how impairing SAA metabolism would promote cell survival (or drives EMT). The authors do make some suggestions in the discussion and I think some insight into this would greatly enhance the study.

Thanks again for this constructive comment.

Firstly, let us clarify that the experiment in Fig. 3c (and 3b) was done for 24 hours. Our results showed that in this experimental time frame, mesenchymal cells (SNU449 and SNU475) stopped growing but did not die, whereas epithelial cells (Huh7, Hep3B, and HepG2) died. These observations are in fact consistent with previous reports that different human cancer cells have varying degrees of methionine dependence (Reference 16, Stern et al., J Cell Physiol, 119, 29-34, 1984). The distinct sensitivities of different HCC lines to SAA depletion prompted us to initiate this project to elucidate the molecular basis underlying these differences. Clearly, SAAs are essential for a number of cellular processes, and we found that extended culture in this methionine/cystine restricted medium leads to the death of all five liver cancer cell lines.

Secondly, the reviewer is right that we still don't have a complete molecular understanding of the mechanism, particularly how rewiring of the SAA metabolism promotes EMT and cell survival. However, as revealed in Fig. 7, 8, and Supplementary Fig. 7, highlighted in Abstract, and discussed in the discussion, our data demonstrate that the defects in CBS-mediated transsulfuration pathway are key for HNF4 α deficiency-induced EMT. We hypothesize that cellular epigenetic programs or signaling pathways regulated by transsulfuration metabolites, particularly Ctt and H₂S, could be involved in HNF4 α -mediated EMT suppression in HCC and we are currently testing this hypothesis. However, delineation of the molecular mechanism and phenotypic significance of different transsulfuration enzymes will require long-term experiments, which we think is beyond the scope of the current study.

Specific points

1. In Figure 4, knock down of HNF4a in HepG2 cells is shown to decrease the expression of the SAA metabolism genes. To support the data in Figure 3 it would be useful to show the expression of these genes in all the HCC cell lines including HNF4a depleted Huh7 and Hep3B cells (which are described in Figure 6E).

Very good points. We have done the requested experiments in Huh7 and Hep3B cells. As shown in new Supplementary Fig. S4d, knocking down HNF4 α in Huh7 and Hep3B cells also reduced the expression of SAA metabolism genes.

S4d

2. The authors show nicely the effect of HNF4a re-expression on reporters in SNU448 cells (Figure 4D) but I think it would be important to show that there is an increase in expression of the endogenous SAA genes in these cells too.

Again, a very good suggestion. We have tested this possibility previously, but the result was negative. Since previous studies have shown that development and progression of HCC is associated with epigenetic silencing of MAT1A, CBS, CDO1¹⁻³, we believe this epigenetic mechanism impedes chromatin accessibility of SAA genes in mesenchymal cells and blocks the binding of HNF4 α in our re-expression experiment.

Per the reviewer's request, we attempted to reverse this epigenetic silencing using DNA methyltransferase (DNMT) inhibitor 5-aza-2'-deoxycytidine (5azaCdR) and/or HDAC inhibitor Trichostatin A (TSA). Our result showed that while 5azaCdR has no impact on HNF4 α -induced expression of SAA genes (not shown), pretreatment with TSA enabled ectopic HNF4 α to reactivate the expression of these endogenous SAA genes in SNU449 cells (new Fig. 4f, right).

3. The analysis of siRNA KD in HepG2 cells is fine, but the authors should also validate the key responses in the other HNF4a expressing cells. As siRNAs can have off target effects (and the authors seem only to use a mixture of 2 siRNAs with no rescue to validate the specificity of the response), a different approach to knock down, such as CRISPR, would also help to strengthen the results.

Thanks for the suggestion. We have shown that knocking down HNF4 α in Huh7 or Hep3B reduces the expression of key SAA genes (new Supplementary Fig. S4d), increases survival during methionine/cystine restriction (Fig. 6e), and enhances resistance to sorafenib (Fig. 6g, for Hep3B). Per the reviewer's request, we also analyzed the impact of HNF4 α knockdown on the expression of EMT markers in the siHNF4 α Huh7 and normal human hepatocytes (new Supplementary Fig. S6a).

Notably, we found that stable knock down of HNF4 α in epithelial liver cancer cells is not a practical strategy for hepatocytes since HNF4 α is required for maintenance of hepatocyte identity. We found that stable knockdown of HNF4 α by shRNAs led to loss of hepatocyte identity and cell senescence. The final selected cell clones, if any, only had a partial reduction (50% at most) of HNF4 α protein (data not shown). After many failed attempts, we settled on knocking down HNF4 α in epithelial HCC cells with siRNA, which results in drastic transient reduction of HNF4 α , as a better experimental model than stable knockdown for the mechanistic experiments in our study. We would like to note that siRNA-mediated knockdown of HNF4 α has been frequently used in studies with HCC or hepatic cells⁴⁻⁸.

As requested by the reviewer, we performed new experiments where we used two different siRNAs targeting HNF4 α separately to confirm the specificity of SAA enzyme downregulation in both HepG2 cells and normal hepatocytes (Fig. 4b and 4d).

4. What happens to the metabolite profiles (as shown in Figure 5) in the SNU448 cells re-expressing HNF4a?

After we worked out the experimental condition (TSA pretreatment) that enables HNF4 α to reactivate SAA genes in SNU449 cells (Fig. 4f), we tried a few times to measure the metabolite profiles in these cells. However, the treated cells were not healthy enough to obtain reliable metabolite data.

5. The use of siRNA treated cells in a 25-day tumorigenesis assay seems difficult – despite the apparent maintenance of reduced mRNA levels in the tumors (Figure S4B). It would appear from Figure 6F that depletion of HNF4a does not impact the survival of HepG2 or Hep3B cells in fed conditions, so it would be possible to generate stable HNF4a depleted lines. I think this needs to be done – at least for the in vivo studies – and would help to validate the in vitro work (see above).

For the reason mentioned in the answer to question 3, we chose to use siRNA treated cells for the tumorigenesis study based on a xenograft mouse model established to generate HNF4 α -deficient tumors^{4,5}. In this model, the authors showed that transient inhibition of HNF4 α by siRNAs initiates a microRNA-inflammatory feedback loop that continuously suppresses HNF4 α .

expression and sustains a stable phenotype of tumorigenesis. In this self-reinforcing circuit, inhibition of HNF4 α , which binds to the promoter of MiR-124, leads to significant reduction of MiR-124 that induces the expression of IL6R and phosphorylation of STAT3, a downstream target of ILR. In turn, enhanced activity of STAT3, which binds to the promoter of MiR-24 and MiR-629, results in upregulation these two MiRNAs that subsequently suppress the expression of HNF4 α . Due to the difficulty to generate stable HNF4 α knockdown cells, as mentioned above, we consider this xenograft model that generates tumors with efficiently suppressed, but not totally depleted HNF4 α , a valid model to study the response of HNF4 α -deficient tumors to methionine restriction. We regret that we did not cite the paper from Hatzia Apostolou et al. to explain the rationale of using siRNA for the xenograft experiment.

Additionally, we would like to point out that stable knockdown of HNF4 α also induces dramatic cell differentiation and senescence (data not shown).

We added a few sentences for our rationale and cited the paper in our revision on Page 15: “Since HNF4 α is required for maintenance of hepatocyte identity⁹⁻¹¹ and stable knockdown of HNF4 α by shRNAs led to loss of hepatocyte identity and cell senescence (data not shown), we chose to use siRNA-treated HepG2 cells for an *in vivo* xenograft experiment based on a xenograft mouse model established to generate HNF4 α -deficient tumors⁴. In this study, the authors demonstrated that siRNA-mediated knockdown of HNF4 α initiates a microRNA-inflammatory feedback loop that continuously suppresses HNF4 α expression and sustains a stable phenotype of tumorigenesis.”

6. Are SNU449 and SNU475 tumors insensitive to methionine restriction?

We asked the same question, and have tried many times to inject SNU449 and SNU475 cells into different types of immunocompromised mice using different injection strategies. However, we found that in our hands, neither NU/J mice (#002019, Jackson Laboratory, the strain we used to generate HepG2 tumors) nor highly immunocompromised NSG mice (#005557, Jackson Laboratory) support formation of tumors from these two mesenchymal cell lines. Accordingly, we were not able to test their sensitivity to methionine restriction *in vivo*.

7. In Figure 7 the authors focus on CBS as a key mediator of the effect of HNF4a on EMT. But what is the effect of CBS depletion on SAA metabolism and the survival of cells under methionine and cysteine depletion? This would seem to be a critical point for the study.

We have already demonstrated that CBS depletion led to altered SAA metabolism, as evidenced by the dramatic reduction of H₂S production (Fig. 5c). CBS knockdown also led to reduced activation of caspase and proapoptotic transcriptional factor *CHOP* (Fig. 7a) and enhanced cell survival (Fig. 7b, previous Fig. S5b) in response to methionine/cystine restriction or sorafenib.

We have discussed some of these details in our discussion part. We modified this part with more details on CBS's impact in H₂S production and cell survival (Page 21): “CBS is tightly regulated by HNF4 α transcriptionally (Fig. 4e, CBS Luc Reporter), and knocking down CBS in HepG2 cells most closely mimics HNF4 α deficiency in promoting the resistance to methionine/cystine restriction and sorafenib (Fig. 7a), inducing mesenchymal morphology (Supplementary Fig. 7b), and reducing H₂S production (Fig. 5c). Moreover, functional restoration of CBS activity in HNF4 α -deficient liver cancer cells, either by supplementation of Ctt

or H₂S (Fig. 7e and 8b) or by overexpression of CBS (Fig. 8c, 8d, and 8e), significantly alleviates stress resistance, cell migration, and EMT induced by HNF4 α deficiency.”

Reviewer #2 (Remarks to the Author); expert on methionine/sulfur metabolism:

The manuscript by Xu et al. investigates a potential liability in some hepatocellular carcinoma (HCC) cells involving sulfur metabolism. It is known that decreased expression of sulfur amino acid (SAA) pathways correlates with increased malignancy in HCC. Based on an analysis of RNAseq data from 373 HCC tumors and 50 normal samples in TCGA, the authors found a positive correlation between HNF-4a expression and expression of SAA metabolism pathways, and a negative correlation between expression of HNF-4a and markers of a mesenchymal phenotype, such as TWIST1. RNAseq data from 25 different established HCC cell lines showed similar correlations. Subsequent investigations were performed on established HCC cell lines, in particular the HNF4a-high expressing cell lines HepG2, HUH7, and Hep3B217, and the HNF4a-low HCC lines SNU449 and SNU475. The authors found that the HNF4a-high lines, in addition to having higher expression of SAA metabolism genes and lower expression of mesenchymal marker genes, had higher sensitivity to Met/Cys restriction, higher basal H₂S production, more of an epithelial phenotype, and higher sensitivity to Sorafenib. ChIP analyses indicate HNF4a interacts directly with 5 genes encoding SAA metabolic functions. Knockdown of HNF4a in HepG2 induced the cells to take on an SNU449-like phenotype, including diminished expression of SAA metabolism genes, diminished expression of epithelial markers, diminished sensitivity to Met/Cys restriction, diminished sensitivity to Sorafenib, increased expression of mesenchymal markers, and increased cell migration. In a mouse xenograft model, HepG2 cells expressing siRNA for HNF4a were less impacted by a Met-restricted diet than were tumors from HepG2 cells expressing a control siRNA. The paper identifies HNF4a as a regulator of SAA metabolism and perhaps also of a mesenchymal phenotype, and the authors conclude HNF4a, via regulating SAA metabolism, determines sensitivity of HCC to Met restriction.

Strengths of the paper.

This is, to my knowledge, the first evidence of HNF4a playing a direct role in regulating the transcription of the reported SAA metabolism genes. The complementary analyses of expression profiles, ChIP data, and knockdown data are compelling. In particular, the correlations between HNF4a and MAT1A, BHMT, and perhaps others in the large TCGA dataset matches pretty well with the ChIP and knockdown data. The robustness of the direct role of HNF4a in this regulation, however, breaks down somewhat in the established cell lines. For example, in comparing the correlation of HNF4a expression and some of its established

targets (HNF1A, Alb) versus to the SAA targets identified in this report (MAT1A, BHMT, CBS - Fig. 1E), there are quite a few unexplained outliers, wherein HNF4a expression patterns match well with characteristics of the established targets, but not the putative SAA targets, in certain cell lines. The presence of outliers, both in the cultures and the TCGA data might be as important as the overall statistical correlations in making generalizations about HCCs.

We are glad to know that the reviewer considers our observations novel and data compelling.

In regard to outliers of SAA targets, we attribute them to the genetic heterogeneity of human liver cancer lines. For example, considering HepG2, Huh7, and Hep3B, even though they are all epithelial liver cancer cells with high expression of HNF4 α , HepG2 has normal expression of wildtype p53, Huh7 overexpresses p53 but with a point-mutated codon, while Hep3B does not express p53 at all ¹². Therefore, although our study uncovers a key role of HNF4 α in regulation of SAA metabolism, it does not exclude the involvement of other factors in this process. It will be interesting to find out if various mutations in different liver cancer cells lines would contribute to distinct SAA metabolism features in outliers.

We add this point in our discussion in the revision (Page 23: “Although our study uncovers a key role of HNF4 α in regulation of SAA metabolism, it does not exclude the involvement of other factors in this process, particularly in human liver cancer cell lines. Human liver cancer lines are genetically highly heterogeneous. For example, considering HepG2, Huh7, and Hep3B, even though they are all epithelial liver cancer cells with high expression of HNF4 α , HepG2 has normal expression of wildtype p53, Huh7 overexpresses p53 but with a point-mutated codon, while Hep3B does not express p53 at all ¹². It will be interesting to find out whether various mutations in different liver cancer cells lines also contribute to distinct SAA metabolism features in future studies.”

Weaknesses of the paper.

Major:

1) Living cells cannot live without a usable source of exogenous sulfur. The cell culture experiments here are reportedly done under conditions of 0 mM Met and 0 mM cystine, which are the only two normal nutritional sulfur amino acids, and are the only two sulfur-containing molecules, in standard DMEM. In Met/Cys-free DMEM, there are no sulfur-containing molecules. Experiments go for up to 24 h under these conditions. Without Met and Cys, not only can there be no protein synthesis, but also no synthesis of any sulfur-containing molecules, including SAM, GSH, CoA, FeS-clusters, and others. Clearly there must have been contaminating sources of sulfur-containing nutrients in the experiments presented, such that the actual conditions were > 0 mM Met/Cys. Possible contaminating sources could have been from small molecules in the serum (Methods do not specify that serum was dialyzed), components of other media supplements, proteolysis of serum proteins, or proteolysis of self via autophagy or similar processes. The actual sources and amounts of nutritional sulfur in these experiments needs to be quantified and presented.

Thanks for this suggestion. We are aware that methionine and cystine are the only two sulfur-containing molecules in the DMEM, and we are also aware that there are other sources of these two SAA in our culture medium. That's the reason we carefully used term “methionine/cystine depletion” but not “methionine/cystine-free” in our original submission, we also stated in the Figure legend (please see page 36, line 8 in the original submission) that “0 μ M” means “no

exogenous Met and Cys”, although the word “exogenous” is still misleading. We apologize for this confusing labeling.

We made our methionine/cystine restricted medium using DMEM without methionine, cystine and glutamine (Sigma, D0422), and supplemented with 2 mM glutamine (ThermoFisher Scientific, 25030081) and 10% dialyzed FBS (10 kD MW cutoff, ThermoFisher Scientific A3382001). We agree with the reviewer that even though the DMEM we used is free of methionine/cystine, binding and transfer proteins such as albumin (66.5 kD), alpha-2-Macroglobulin (720 kD), hemoglobin (64 kD)¹³ remain in the dialyzed FBS and can be lysed into small peptides or single amino acids by cellular proteases. Therefore, there must be a sufficient amount of methionine/cystine derived from proteolysis of large serum proteins, or from proteolysis of cell components via autophagy or similar processes, to sustain the viability of the cells. This is particularly important for mesenchymal liver cancer cells treated with methionine restriction for 24 hours.

As requested, we quantified the actual concentration of sulfur-containing compounds in our culture media before and 24 hours after cell culture by LC-MS. Our results showed that our complete medium contains 130 μM methionine and 160 μM cystine, and the methionine/cystine restricted medium contains 0.12 μM methionine and undetectable levels of cystine. Further LC-MS analysis revealed that the concentrations of methionine and cystine are 139 μM and 200 μM , respectively, in complete medium, and 0.15 μM and 0.11 μM , respectively, in restricted medium after 24-hour cell culture, indicating that both media are able to maintain or even increase their respective methionine/cystine concentrations during this experimental timeframe. This observation is consistent with the notion that small peptides or single amino acids can be derived from proteolysis of large serum proteins, or from proteolysis of cell components via autophagy or similar processes during cell culture. We added this information on Page 9. Based on these measurements, we changed “methionine/cystine depleted medium” into “methionine/cystine restricted medium (MCR)” in our revised manuscript. We also corrected concentration labels in all Figures.

2) Met is essential (not mentioned in the paper). Regardless of cystine availability, human cells cannot survive without an exogenous source of Met. Again, the conditions in this study cannot truly be 0 mM Met/Cys, as reported. Interestingly, Thr is also essential (also not mentioned), and Fig. 3D shows the expected lethality under Thr-restriction. This even further confounds comprehension of what source of Met is rescuing the cells in this study, as proteolysis or autophagy would each yield Met and Thr.

First of all, we did mention that methionine as a sulfur-containing essential amino acid in the first paragraph of Introduction (Page 3 in the original submission, “For example, restriction of dietary methionine, a sulfur-containing essential amino acid enriched in animal products, has been shown to suppress proliferation and progression of a variety of tumors, including colon, prostate, and breast cancer¹⁴⁻¹⁹.”). Therefore, we are well aware that human cells cannot survive long without an exogenous source of Met. This is precisely the reason why methionine/cystine restriction for our gene expression experiments only lasts for 6 hours, and for the survival/apoptosis experiments the restriction is only for 24 hours. Please note that the conditions we used for methionine restriction are standard in the field and have been adopted by various studies using amino acid deprivation²⁰⁻²².

Secondly, we would like to emphasize that our major message here is “mesenchymal liver cancer cells are more resistant to methionine/cystine restriction than epithelial liver cancer

cells". We measured this resistance by their insensitivity to acute methionine/cystine restriction (6 hours)-induced AAR (Fig. 3a) and delayed death in response to 24-hour methionine/cystine restriction-induced nutritional stress (Fig. 3b and 3c). Our results indicate that epithelial and mesenchymal liver cancer lines have distinct sensitivities to methionine/cystine restriction, but do not suggest that Met is not essential to some cells. Again, as stated in our answer to the point 1), we agree with the reviewer that our condition is not truly 0 μ M methionine/cystine, even regarding the exogenous source. We measured the actual concentrations of methionine/cystine in our medium by LC-MS, which are 130/160 μ M for the complete medium and 0.12/0 μ M for the methionine/cystine restricted medium, respectively. We corrected concentration labels in the Figures.

Thirdly, we apologize that we did not mention that Thr (and Leu) are also essential, and Glu is conditionally essential—we added this information in our revision on Page 10. Again, we are well aware of the essential (or conditionally essential) nature of these amino acids. The essential (or conditionally essential) nature of these amino acids is the exact reason why we used them as controls to show that the differential responses of epithelial and mesenchymal liver cancer cells to methionine/cystine restriction is specific, rather than a general sensitivity to depletion of any essential (or conditionally essential) amino acid. We added a sentence to make this point clear in the revision on Page 10-11: "This observation suggests that additional mechanisms are involved in the differential responses of epithelial and mesenchymal liver cancer cells to methionine/cystine restriction, not simply because methionine is essential and indispensable for protein synthesis."

Finally, we believe the reviewer was surprised by our observation that restriction of an essential amino acid in culture medium does not lead to cell death. Our answer here is the treatment timeframes. Given enough time, depletion of an essential amino acid will indeed lead to the death of all human cells. However, our data here indicate that different essential amino acid depletions have different kinetics in killing a specific cell type, and distinct cell types have distinct response kinetics in response to the same amino acid depletion. For instance, with the 24-hour timeframe, the sensitivities of epithelial liver cancer cells to amino acid depletion is Thr > Met/Cys > Leu, Glu. On the other hand, the sensitivities of mesenchymal liver cancer cells is Thr > Met/Cys, Leu, Glu (Fig. 3d). We believe these distinct sensitivities to the depletion of different essential amino acids reflects the distinct requirements of each cell line for the different essential amino acids. According to our observations, it appears that both epithelial and mesenchymal liver cancer cells have high demand for Thr, but mesenchymal liver cancer lines can survive at lower Met levels compared to epithelial lines. Based on our new LC-MS data, it is possible that 0.12-0.15 μ M Met concentration in our MCR medium is above the viability threshold of mesenchymal lines but below the threshold of the epithelial lines for the 24-hour timeframe. In contrast, the viability threshold for Thr is higher for both classes of cells and the amount produced by proteolysis or autophagy is insufficient to maintain the viability of both epithelial and mesenchymal lines.

3) Elimination of the Cys source along with Met from the medium was not justified, since having a Cys source cannot provide Met. Would similar results be obtained with only Met restricted?

As stated in our main text (Page 9), the reason for the elimination of both Cys and Met from the medium is that the commonly used animal dietary "methionine restriction" regimen restricts methionine in the absence of cystine. This dietary methionine/cystine intervention has been shown to extend life span, delay aging, prevent metabolic diseases, and more importantly for our studies, reduce cancer growth, and sensitize cancer cells to chemotherapy and radiation in

mice²³⁻²⁶ (References 51, 52, 15 and 8 in the original submission). We modified this sentence to clarify our justification.

Per the reviewer's request, we tested whether Met only restriction will lead to similar results in liver cancer cells. Our result in Fig. 7f showed that methionine only restriction (MCR + Cystine) also induce cell death in control HepG2 cells, but to a significantly smaller extent compared to methionine/cystine restriction (siNeg HepG2). Therefore, elimination of cystine does contribute to traditional "methionine restriction"-induced impact. On the other hand, it is interesting to note that methionine only restriction (MCR + Cystine) has the same impact as MCR on cell survival in HNF4 α -deficient cells (siHNF4 α HepG2 cells and HNF4 α -negative SNU449 cells), suggesting that the impact of cystine restriction is dependent on HNF4 α . Based on our present study, the primary impact of the traditional methionine restriction regimen (that also eliminates cystine source) on liver cancer cells is mediated by HNF4 α -regulated transsulfuration, which involves both methionine and cystine metabolism.

4) All HNF4a-high cell lines were grown in DMEM + 10% FCS; all HNF4a-low cell lines were grown in RPMI + 10% heat-inactivated FCS. Experiments were done in amino acid-defined DMEM + 10% FCS. Therefore, in the experiments, the HNF4a-low, but not the -high, cell lines were exposed to media conditions different from what they are adapted to.

Good point. We used standard protocols to maintain the different liver cancer cell lines. We have noticed that DMEM contains 200 μ M of methionine while RPMI contains 100 μ M of methionine, but both media contain 200 μ M of cystine. We have tested (not shown in the paper) that culturing SNU449 and SNU475 in DMEM+ 10% FBS did not affect the expression levels of HNF4A and SAA genes in these cells. Therefore, it is highly unlikely that media change contributes to the differential responses of the mesenchymal lines to methionine/cystine depletion.

Importantly, our siHNF4A experiments in HepG2 (siNeg and siHNF4 α HepG2 cells were cultured in the same DMEM-based medium) and HNF4 α overexpression in SNU449 cells (again, pcDNA3 and pcDNA3-HNF4 α SNU449 cells were cultured in the same RPMI medium) further confirmed that their distinct SAA gene expression and EMT features are primarily determined by the HNF4 α status, rather than their culture conditions.

5) A mechanistic connection between the SAA metabolism pathways and sorafenib is not discussed in the paper, making it difficult to follow the logic behind why a kinase inhibitor should interact with Met restriction.

There are two reasons for us to test sorafenib.

First, we stated in the main text that sorafenib functions as an inhibitor of the cystine-glutamate antiporter (system x_c⁻) in addition to multiple kinases^{27, 28} (References 55 and 56 in the original submission, Page 10-11). Thus, sorafenib treatment serves as another way to deplete cells of cystine. This is the direct connection between SAA metabolism and sorafenib.

Second, sorafenib is one of the very few FDA-approved drugs that show some limited efficacy in advance HCC. Our data in Fig. 3e showed that methionine restriction enhances the effect of sorafenib on epithelial cells but not on mesenchymal cells, suggesting that dietary methionine restriction may be useful strategy to sensitize epithelial liver tumors to sorafenib treatment. This may have important clinical implications.

6) The conclusions of HNF4a-high and -low cell lines having different susceptibilities to sorafenib under Met/Cys restriction is not supported by data. Fig. 3E shows that the biggest difference in survival occurs at 0 μM sorafenib. As doses of sorafenib increase, differences between Met/Cys restricted and Met restored conditions diminish for all cell lines.

From our data in Fig. 3e, it is clear that when given as sole treatment, sorafenib had an IC₅₀ of more than 10 μM for mesenchymal cells (SNU449 and SNU475), but an IC₅₀ of 2.5 to 5 μM for all three epithelial cells (Huh7, Hep3B, and HepG2). So HNF4 α -high cell lines have increased sensitivity to sorafenib than HNF4 α -low cells in complete medium. We do agree with the reviewer that methionine/cystine restriction introduces the biggest differences in survival of HNF4 α -high epithelial cells when there is no sorafenib, and increasing sorafenib concentration diminishes this difference. However, if we consider that sorafenib functions as an inhibitor of the cystine-glutamate antiporter (system x_c⁻)^{27, 28} in addition to multiple kinases (as mentioned in our response to the Question 5), this observation makes sense. The effect of sorafenib on cell survival is partly through prevention of cystine import from the medium, thus, it is reasonable that a part of its function is not manifested when there is no cystine in the medium. In other words, the effects of methionine/cystine restriction and sorafenib are not synergistic but less than additive in our methionine/cystine restriction condition.

Nevertheless, based on our data in Fig. 3e, it appeared that sorafenib still has significantly lower IC₅₀ in both Huh7 and Hep3B cells (~1-3 μM) than in SNU449 and SNU475 cells (from 10 to more than 20 μM) in the methionine/cystine restriction condition. The only exception is HepG2 cells, in which methionine/cystine restriction increased the IC₅₀ of sorafenib from ~5 μM to ~15 μM .

7) The impacts of Met/Cys restriction on cellular levels of Met, Cys, or GSH, and on protein

Figure S3

synthesis rates, proliferation, or oxidative stress, are not shown in the cell culture models.

As requested by the reviewer, we performed new experiments and found that methionine/cystine restriction quickly depletes intracellular levels of

Met and Cys, and reduces GSH in both HepG2 cells and SNU449 cells (new Supplementary Fig. S3).

We also performed new experiments to assess ROS, proliferation and protein synthesis. Methionine/cystine restriction elevated intracellular ROS levels in control HepG2 cells, but not in siHNF4 α HepG2 cells nor SNU449 cells (Fig. S5b). On the other hand, this intervention was able to reduce the fraction of S-phase cells and induce G1 arrest (Fig. S5c) and reduce protein synthesis (Fig. S5d) in all cells with different status of HNF4 α . In complete medium, protein synthesis rate is much lower in HNF4-depleted or deficient cells, proliferation is slower, and level of ROS is higher. Taken together with the upregulation of AAR response genes (Fig 3a), these results suggest that HNF4-depleted or deficient cells are under constant chronic stress that may activate resistance mechanisms, including EMT, which prepare the cells to better cope with further insults such as MCR or sorafenib^{29, 30}.

8) The authors do not show that Met-restriction in the xenograft models affects the amount of available Met in circulation in the animals. The condition used (chow containing 0.172% Met) is only 2- to 4-fold below Met levels in many standard rodent chow formulations. Additional Met might come from gut microbes. The mouse studies should be supported by measurements of Met in circulation and in tissues.

The control diet (0.86% Met) and MCR diet (0.172% Met) we used in our xenograft experiments are special diets purchased from Dyets (they are not chow diet containing 0.172% Met). A recent study from our co-authors, Xia Gao and Jason Locasale²⁵ (Reference 15 in the main text), has measured circulating and tissue Met levels for animals under control and MCR diets, which are comparable to the diets used in the current study. They showed that the MCR diet reduces the levels of plasma methionine (to ~50% of those in chow diet fed mice) within two days, and these levels are sustained throughout the treatment (Fig. 1d in this paper). They also showed that the MCR diet alters methionine metabolism in their colorectal PDX tumors and liver tissues after feeding (Extended Data Fig. 2 in this paper).

We added a sentence on Page 16 to clarify this point: “Similar MCR diet has been recently shown to reduce plasma methionine levels and alter methionine metabolism in colorectal PDX tumors and liver tissues in mice²⁵.”

Minor weaknesses

1) It is well established that phenotypically more differentiated HCCs are less malignant and less metastatic. HNF4a is a major determinant of the differentiated state of hepatocytes and of hepatocellular carcinomas. Other physical and molecular characteristics of differentiated hepatocytes include (1) strong expression of hepatocyte-specific genes (e.g.s Alb and HNF1A) and many SAA metabolism genes (MAT1A, BHMT, CBS, CTH); and (2) an epithelial morphology. Although most correlations between SAA metabolism, malignancy, metastasis, and HNF4a expression reported here are neither new nor surprising, it is novel and satisfying that the presented study suggests a direct regulatory connection between HNF4a and both the upregulation of SAA metabolism genes and the epithelial phenotype. It is, however, an unnecessary confusion for the field for this study to re-name “differentiated” and “undifferentiated” HCCs as “epithelial” and “mesenchymal”, respectively.

Thanks for the positive comments on the novelty of this study.

Both “differentiated” and “undifferentiated” HCCs and “epithelial” and “mesenchymal” are used in the field by different studies ³¹. We used “epithelial” and “mesenchymal” to emphasize the important roles of HNF4 α and SAA metabolism enzymes in epithelial to mesenchymal transition (EMT).

2) Cysteine and cystine are confused at several places in the text.

We apologize for this overlook. We fixed a number of places, in which cystine was confused with cysteine, including methionine/cystine restricted medium and diets. Also, System Xc is a cystine-glutamate antiporter not a “cysteine-glutamate antiporter”. “Cysteine” was only used for intracellular concentration of cysteine, which is the reduced form of this amino acid.

3) “Relative enrichment” (or similar) on the Y-axis of several figures is not defined. Since the scales start at 0, it suggests this is a scale for which a zero-value is possible (i.e., not a ratio or fold-change).

Thanks for pointing this out. We modified the figure and used the % of Input as the Y-axis (Fig. 4a).

4) Figs 2A and 5G are unclear. “Impact” is not defined, nor are there descriptions of what is signified by dot-color, -intensity, or -size.

We apologize for the lack of detailed descriptions. These two graphs were automatically generated by the Pathway Analysis module of MetaboAnalyst 4.0., where the Y axis is the enrichment p values and the X axis is the pathway impact values, indicative of the centrality and enrichment of a pathway. The color of a circle is indicative of the level of enrichment significance, with yellow being low and red being high. The size of a circle is proportional to the pathway impact value of the pathway.

We added the following explanation in the figure legend of Fig. 2a: “The 174 metabolites that displayed significantly different abundances between SNU449 and HepG2 cells were subjected for the pathway enrichment analysis (Y axis, enrichment p values) and the pathway topology analysis (X axis, pathway impact values, indicative of the centrality and enrichment of a pathway) in the Pathway Analysis module of MetaboAnalyst 4.0 (n=3 repeats, $p < 0.05$, $|FC| > 1.5$). The color of a circle is indicative of the level of enrichment significance, with yellow being low and red being high. The size of a circle is proportional to the pathway impact value of the pathway”. A similar description was also added in the legend of Fig. 5g.

5) ATCC describes HepG2 cells as “non-tumorigenic in immune compromised mice”; yet xenograft experiments presented in Fig. 6 contradict this. High quality histology of the tumors is not shown. Markers do not validate the origins of the tissue shown. Some further explanation to resolve this discrepancy would be helpful.

ATCC describes HepG2 cells as “non-tumorigenic in immune SUPPRESSED mice”. However, HepG2 xenograft has been reported in several published studies ^{4,33}. Obviously, we detected xenograft tumor formation of HepG2 cells in our experiment. It is possible that HepG2 are non-tumorigenic when relatively low cell number is injected (1×10^6), but become tumorigenic with higher inoculum ($>5 \times 10^6$). We inoculated 5×10^6 in our xenograft experiments. As requested by the reviewer, we added H&E staining images of xenografted tumors (new Supplementary S6d).

Reviewer #3 (Remarks to the Author); expert on HCC and transcription:

Comments for the authors

The authors aimed to identify importance of HNF4A-SAA metabolism axis involved in methionine restriction in HCC by bioinformatics analyses, metabolomics, and molecular and cellular characterizations. Expression of HNF4A was reduced in HCC patients in TCGA dataset, positive correlation between HNF4A and SAA enzymes was also found, and low expression of key SAA enzymes showed poor prognosis in HCC patients. Similar results were observed in HCC or hepatoblastoma cell lines, and HNF4A was found to directly regulates expression of SAA enzymes. Suppressed expression of HNF4A or SAA enzymes in HNF4A-positive HepG2 cells showed increased resistance to methionine restriction and sorafenib, and induced cell migration and expression of mesenchymal markers. These data indicate that HNF4A-SAA metabolism axis play an important role to determine the sensitivity of HCC to methionine restriction. **This is very interesting study**, but the authors should explain, discuss, and/or demonstrate the following points to strength the manuscript.

Thanks for the positive comments on the study.

Major

(1) Figs. 1B and C; positive correlation between HNF4A and SAA enzymes is also found in both viral-HCC and non-viral HCC?

The answer is yes. We divided the HCC patients in the TCGA LIHC dataset into patients without viral hepatitis serologies (n=207) and patients with any viral hepatitis serology, including positive of hepatitis c antibody, hepatitis b surface antigen, hepatitis b surface antibody, hepatitis b core

antibody, and/or hepatitis b DNA (n=154). The correlation results are shown in new Supplementary Fig. S2a and S2b.

(2) Fig. 1D; negative correlation between SAA enzymes and TWIST1 was found, but the authors do not use TWIST1 as an EMT marker in further Figures (Figs. 6A, 7E, and S5C). Why?

Thanks for pointing this out. As suggested, we analyzed TWIST1 expression in the following new Figures: Fig. 6a, 8a, 8b, 8e, and S6a.

(3) The authors mainly use HepG2 cells as HCC cells. However, HepG2 cells are hepatoblastoma cells, but not HCC cells (Lopez-Terrada, D et al, Human Pathol, 40, 1512-1515, 2009). In addition, protein expression of CBS is almost same between HepG2 and SNU449 (Fig. 1G). Protein expression of HNF4A and most SAA enzymes is highest in Huh7 and lowest in SNU475 among five cell lines. The authors should mainly use Huh7 and SNU475 as epithelial and mesenchymal cells, respectively. Otherwise, the authors should use both HepG2 and Huh7 as epithelial cells, and both SNU475 and SNU449 as mesenchymal cells to prove generality.

Thanks for pointing this out. Even though the paper cited was published more than 10 years ago, HepG2 is still recognized as HCC cells in the ATCC website <https://www.atcc.org/products/all/HB-8065.aspx#characteristics> with the description “In addition to the hepatocellular carcinoma or hepatoma cell line based on the original publication (PubMed: 6248960), HepG2 is also referred as hepatoblastoma cell line (PubMed: 19751877).”

With that, we agree with the reviewer that we need to use some additional HCC lines to verify our conclusion, and simply refer HepG2 as an HCC line will cause confusion. To address these concerns, we made the following changes: (1) We replaced “hepatocellular carcinoma” or “HCC” in the manuscript title and many other places in the manuscript with “liver cancer” or

“liver cancer cells”. (2) we extended our study to additional HCC lines, including Huh7 and Hep3B, as well as normal human hepatocytes. Our current data showed that knocking down HNF4 α in Huh7, Hep3B, or normal human hepatocytes reduces the expression of key SAA genes (new Fig. S4d and 4d), increases resistance to methionine/cystine restriction (Fig. 6e) and sorafenib (Fig. 6f for Huh7 and Fig. 6g for Hep3B), and elevates the expression of EMT markers (Supplementary Fig. 6a).

(4) Fig. 3E; why did Met/Cys-depleted HepG2 cells show resistance to cell death induced by sorafenib compared to Huh7 and Hep3B cells? Because of hepatoblastoma cells?

There are some intrinsic differences among these three epithelial lines even though they share a lot of common features and are clustered together as shown in Fig. 1e. It is possible that these additional intrinsic genetic differences contribute to their differential response to sorafenib. Therefore, although our study uncovers a key role of HNF4 α in regulation of SAA metabolism, it does not exclude the involvement of other factors in this process. It will be interesting to find out if various mutations in different HCC cells lines would contribute to distinct SAA metabolism features in outliers.

We added a paragraph at the end of Page 23 to discuss this point in the revision: “Although our study uncovers a key role of HNF4 α in regulation of SAA metabolism, it does not exclude the involvement of other factors in this process, particularly in human liver cancer cell lines. Human liver cancer lines are genetically highly heterogeneous. For example, considering HepG2, Huh7, and Hep3B, even though they are all epithelial liver cancer cells with high expression of HNF4 α , HepG2 cells have normal expression of wildtype p53, Huh7 cells overexpress a mutant p53, while Hep3B cells are p53 null¹². It will be interesting to find out whether various mutations in different liver cancer cells lines also contribute to distinct SAA metabolism features in future studies.”

(5) Fig. 4A; the authors should show CHIP-IgG of SNU449.

The reason why we did not include IgG control is because ChIP-HNF4 α in SNU449 was used as an additional negative control for ChIP-HNF4 α in HepG2 cells, as this cell line does not express HNF4 α . To clarify this point we added the following sentence to the legend of Fig. 4a: “IgG ChIP in HepG2 cells and anti-HNF α ChIP in HNF4 α -negative SNU449 cells serve as negative controls”.

(6) Fig. 4D; luc activity of wild-type MAT1A promoter was induced about 2.5-fold by HNF4A, but the mutated promoter was also induced about 2.5-fold by HNF4A, meaning that the mutated HNF4A bind site is not functional. Are there any functional HNF4A binding sites in the MAT1A promoter as shown in Fig. S3? Also, expression of MAT1A, BHMT, and CBS mRNA and/or protein is induced by HNF4A in SNU449 cells?

Good points. There are other HNF4 α binding sites on this promoter, but the one we mutated (from gcttcagagtca to gcgggggggggga) is the best match to the consensus motif of HNF4 α binding sites. We mutated the same site into another mutation (gcttttttttga) for the revision and the result from this new mutant Luc reporter is shown in new Fig. 4e.

Analyzing the endogenous MAT1A, BHMT, and CBS mRNA and/or protein after HNF4 α re-expression is also a very good suggestion. We have tested this possibility previously, but the result was negative. Since previous studies have shown that development and progression of HCC is associated with epigenetic silencing of MAT1A, CBS, CDO1¹⁻³, we believe this epigenetic mechanism impedes chromatin accessibility of SAA genes in mesenchymal cells and blocks the binding of HNF4 α in our re-expression experiment.

Per the reviewer’s request, we attempted to reverse this epigenetic silencing using DNA methyltransferase (DNMT) inhibitor 5-aza-2’-deoxycytidine (5azaCdR) and/or HDAC inhibitor Trichostatin A (TSA). Our result showed that while 5azaCdR has no impact on HNF4 α -induced expression of SAA genes (not shown), pretreatment with TSA enabled ectopic HNF4 α to reactivate the expression of these endogenous SAA genes in SNU449 cells (new Fig. 4f, right).

(7) Fig. 6A; the authors should investigate expression of epithelial and mesenchymal markers, cellular morphology, cell migration, and apoptosis in HNF4A-overexpressed mesenchymal cells.

Thanks for the suggestion. We have shown in Fig. 7g in the original submission that re-expression of HNF4 α in SNU449 decreased cell migration. The new data (new Fig. 4f) showed that after pre-treatment with TSA, overexpressed HNF4 α is able to reactivate the expression of SAA genes in SNU449 cells. Since cells are not in a healthy state after TSA treatment, we did not get reliable data for cellular morphology and apoptosis.

The impact of re-expression of HNF4 α in SNU449 on the expression of EMT markers is also complicated. As shown in the Figure below, under normal condition, expression of HNF4 α in SNU449 cells only increased the expression of one epithelial marker, CPED1 (HNF4 α DMSO). However, when combined with TSA treatment, expression of HNF4 α surprisingly increased not only the expression of epithelial markers but also many tested mesenchymal markers (HNF4 α TSA). This is, possibly, due to the fact that TSA alone could increase the expression of these mesenchymal markers (pcDNA3 TSA), which has also been reported in the literature^{34, 35}.

(8) Fig. 6G; expression of HNF4A in Hep3B is much lower compared to HepG2 and Huh7. Why did the authors use Hep3B, but not Huh7? In Fig. 1G, expression of HNF4A in Huh7 is higher than that in HepG2, but why is expression of HNF4A in Huh7 lower than that in HepG2 (Fig. 6E)?

There are no particular reasons for using Hep3B in our study. Since we have already shown that knocking down HNF4 α in HepG2 leads to resistance to both methionine/cystine restriction and sorafenib treatment (original Fig. 6f, new Supplementary Fig. 6b), we just wanted to confirm this observation in another epithelial liver cancer cell line.

Per the reviewer's request, we confirmed this result in Huh7 for the revision, and the data is shown in new Fig. 6f.

Thanks for pointing out the discrepancy in Fig. 1g and 6e. This is a typo, Huh7 and HepG2 were accidentally switched in Fig. 6e. We corrected it in the revision.

(9) Fig. 6H and I; there was significant difference in Ki67+ area between siHNF4A MR and siNeg MR groups (Fig. 6I), but were there significant difference in tumor volume between siHNF4A MR and siNegMR between these groups (Fig. 6H)?

Yes, at the final time point, the tumor volume difference between siHNF4 α MCR and siNeg MCR is significant ($p= 0.0079$ with Mann-Whitney test). We labeled this in the revision.

(10) Fig. 7B; the authors mentioned that knockdown of CBS and CDO1 increased stress resistance and induced cell migration. Did knockdown of each SAA enzyme enhance mesenchymal cellular morphology as shown in Fig. 6B?

The answer is yes. Please see the morphology of HepG2 cells with or without knockdown of HNF4 α or individual SAA enzymes (Fig. S7b). It is evident that knockdown of CBS or CDO1 induces mesenchymal cellular morphology in a similar way to HNF4 α knockdown.

(11) Fig. 7E; why did knockdown of CBS elevate expression of CDH1 in spite of elevated expression of several mesenchymal markers? The authors should show whether enhanced mesenchymal cellular morphology does occur in CBS-knockdowned HepG2 cells.

Good point. Although CDH1 has been widely used as an epithelial marker associated with better prognosis, emerging data show that CDH1 may also facilitate metastasis in certain cases ³⁶.

Please see above Supplementary Fig. S7b in our response to point #10, knocking down CBS induced mesenchymal cellular morphology.

(12) Overexpression of individual SAA enzyme suppressed EMT in SNU449?

Overexpression of individual SAA enzyme significantly reduced the mesenchymal characteristics of SNU449 cells (Fig. 8e). Overexpressing CBS in SNU449 cells primarily suppressed the expression of a number of mesenchymal markers that were induced in siHNF4 α HepG2 cells, including *VIM*, *CDH2*, *ITGAV*, *COL3A1*, *COL4A5*, *COL6A1*, and *TWIST1*. Overexpression of BHMT not only repressed the expression of many mesenchymal markers but also induced the expression of two epithelial markers that were repressed in siHNF4 α cells.

(13) The authors should show whether individual SAA enzyme is a direct target of HNF4A using human normal primary hepatocytes.

Thanks for the suggestion. We identified a source of human normal primary hepatocytes and showed that knocking down HNF4 α in these cells also significantly reduced the expression of all tested SAA genes (Fig. 4d).

S4d

(14) It is not clear whether SAA enzymes such as CBS is involved in HCC progression because this research is mainly based on data using epithelial and mesenchymal cell lines. Thus, it is important to investigate whether deficiency of HNF4A-SAA metabolism axis is critical for HCC progression using HCC model experiments using wild-type mice, liver-specific Hnf4a-null mice, and/or SAA enzyme (especially CBS)-deficient mice.

This is a great suggestion. We have also considered the suggested experiments. However, literature reports indicate that both liver-specific Hnf4 α -null mice³⁷ and CBS-deficient mice³⁸ die at young age, while HCC models in mice (DEN-High fat diet model, for example) take 8-9 months to establish. Therefore, these experiments are technically not feasible.

Minor

(1) Fig. S1A; expression of BHMT2 is significantly suppressed in HCC compared to normal liver, but there's no description of BHMT2 hereafter. BHMT2 is not important for SAA metabolism in liver compared to BHMT?

According to Li et al.³⁹, although BHMT2 mRNA has been observed in human liver and kidney, BHMT2 protein is unstable. Therefore, BHMT is the major functional enzyme in the liver.

(2) Figs. S1B and S2B legends; there's no explanation of blue and red bars.

We apologize for this omission. The blue line is the survival probability of patients with low expression (bottom 33%) of indicated gene, and the red line is the survival probability of patients with high expression (top 33%) of indicated gene. We added the explanation for the blue and red lines in the revision.

(3) Fig. 3A; there's no explanation of blue bar.

Again, we apologize for this omission. The blue bars indicate two mesenchymal liver cancer lines.

(4) Line 189; "...displayed elevated basal levels of many of these genes..." should be modified as follows. "...displayed elevated basal levels of ATF4 and CHOP..."

Thanks for the suggestion, we made the change as suggested.

(5) Fig. S3A-C legend; the authors should provide a detailed description.

Thanks for the suggestion, we added the following information at the Legend for this figure (new supplementary Fig. 4a-4c) “Publicly available data for the HNF4 α ChIP-seq from human liver or HepG2 cells from the ENCODE project (PMID:29126249) were displayed in UCSC genome browser for MAT1A (a), BHMT (b), and CBS (c) genes, and the enrichment scores and consensus HNF4 α binding sites are shown for peaks indicated by the red arrows. Blue arrows indicate additional ChIPseq peaks. Red bars: promoter regions cloned and mutated in the luciferase assay.”

(6) Line 265; “49 metabolites” should be corrected to “48 metabolites”?

Yes, it should be 48 metabolites. But please note that we re-analyzed our metabolomic datasets for the revision using a stringer cutoff ($|FC| > 1.5$ instead of 1.3), so this number is changed to 34 in the revision.

References:

1. Zhang, Y. *et al.* Hydrogen sulfide, the next potent preventive and therapeutic agent in aging and age-associated diseases. *Mol Cell Biol* **33**, 1104-1113 (2013).
2. Zhang, Z., Xu, L. & Sun, C. Comprehensive characterization of cancer genes in hepatocellular carcinoma genomes. *Oncol Lett* **15**, 1503-1510 (2018).
3. Choi, J.I. *et al.* Promoter methylation of cysteine dioxygenase type 1: gene silencing and tumorigenesis in hepatocellular carcinoma. *Ann Hepatobiliary Pancreat Surg* **21**, 181-187 (2017).
4. Hatzia Apostolou, M. *et al.* An HNF4 α -miRNA inflammatory feedback circuit regulates hepatocellular oncogenesis. *Cell* **147**, 1233-1247 (2011).
5. Cairo, S. & Buendia, M.A. How transient becomes stable: an epigenetic switch linking liver inflammation and tumorigenesis. *J Hepatol* **57**, 910-912 (2012).
6. Morimoto, A. *et al.* An HNF4 α -microRNA-194/192 signaling axis maintains hepatic cell function. *J Biol Chem* **292**, 10574-10585 (2017).
7. Taniguchi, H. *et al.* Loss-of-function mutations in Zn-finger DNA-binding domain of HNF4A cause aberrant transcriptional regulation in liver cancer. *Oncotarget* **9**, 26144-26156 (2018).
8. Burdin, D.V. *et al.* Diabetes-linked transcription factor HNF4 α regulates metabolism of endogenous methylarginines and beta-aminoisobutyric acid by controlling expression of alanine-glyoxylate aminotransferase 2. *Sci Rep* **6**, 35503 (2016).
9. Sladek, F.M., Zhong, W.M., Lai, E. & Darnell, J.E., Jr. Liver-enriched transcription factor HNF-4 is a novel member of the steroid hormone receptor superfamily. *Genes Dev* **4**, 2353-2365 (1990).
10. Sladek, F.M. & Seidel, S.D. Hepatocyte Nuclear Factor 4 α . *Nuclear Receptors and Genetic Diseases*, 309-361 (2001).
11. Bolotin, E., Schnabl, J. & Sladek, F.M. HNF4A (Homo sapiens) Transcription Factor Encyclopedia. <http://www.cisreg.ca/tfe> (2010).
12. Gomes, A.R. *et al.* Influence of P53 on the radiotherapy response of hepatocellular carcinoma. *Clin Mol Hepatol* **21**, 257-267 (2015).
13. Yang, Z.Q. & Xiong, X.R. Culture Conditions and Types of Growth Media for Mammalian Cells. *INTECH DOI: 10.5772/52301* (2012).
14. Guo, H. *et al.* Therapeutic tumor-specific cell cycle block induced by methionine starvation in vivo. *Cancer Res* **53**, 5676-5679 (1993).

15. Poirson-Bichat, F., Gonfalone, G., Bras-Goncalves, R.A., Dutrillaux, B. & Poupon, M.F. Growth of methionine-dependent human prostate cancer (PC-3) is inhibited by ethionine combined with methionine starvation. *Br J Cancer* **75**, 1605-1612 (1997).
16. Komninou, D., Leutzinger, Y., Reddy, B.S. & Richie, J.P., Jr. Methionine restriction inhibits colon carcinogenesis. *Nutr Cancer* **54**, 202-208 (2006).
17. Sinha, R. *et al.* Dietary methionine restriction inhibits prostatic intraepithelial neoplasia in TRAMP mice. *Prostate* **74**, 1663-1673 (2014).
18. Hens, J.R. *et al.* Methionine-restricted diet inhibits growth of MCF10AT1-derived mammary tumors by increasing cell cycle inhibitors in athymic nude mice. *BMC Cancer* **16**, 349 (2016).
19. Ables, G.P. & Johnson, J.E. Pleiotropic responses to methionine restriction. *Exp Gerontol* **94**, 83-88 (2017).
20. Longchamp, A. *et al.* Amino Acid Restriction Triggers Angiogenesis via GCN2/ATF4 Regulation of VEGF and H2S Production. *Cell* **173**, 117-129 e114 (2018).
21. Tang, X. *et al.* Comprehensive profiling of amino acid response uncovers unique methionine-deprived response dependent on intact creatine biosynthesis. *PLoS Genet* **11**, e1005158 (2015).
22. Chen, H., Pan, Y.X., Dudenhausen, E.E. & Kilberg, M.S. Amino acid deprivation induces the transcription rate of the human asparagine synthetase gene through a timed program of expression and promoter binding of nutrient-responsive basic region/leucine zipper transcription factors as well as localized histone acetylation. *J Biol Chem* **279**, 50829-50839 (2004).
23. Orentreich, N., Matias, J.R., DeFelice, A. & Zimmerman, J.A. Low methionine ingestion by rats extends life span. *J Nutr* **123**, 269-274 (1993).
24. Miller, R.A. *et al.* Methionine-deficient diet extends mouse lifespan, slows immune and lens aging, alters glucose, T4, IGF-I and insulin levels, and increases hepatocyte MIF levels and stress resistance. *Aging Cell* **4**, 119-125 (2005).
25. Gao, X. *et al.* Dietary methionine influences therapy in mouse cancer models and alters human metabolism. *Nature* **572**, 397-401 (2019).
26. Sanderson, S.M., Gao, X., Dai, Z. & Locasale, J.W. Methionine metabolism in health and cancer: a nexus of diet and precision medicine. *Nat Rev Cancer* **19**, 625-637 (2019).
27. Dixon, S.J. *et al.* Pharmacological inhibition of cystine-glutamate exchange induces endoplasmic reticulum stress and ferroptosis. *Elife* **3**, e02523 (2014).
28. Yang, W.S. & Stockwell, B.R. Ferroptosis: Death by Lipid Peroxidation. *Trends Cell Biol* **26**, 165-176 (2016).
29. Jiang, J. *et al.* Redox regulation in tumor cell epithelial-mesenchymal transition: molecular basis and therapeutic strategy. *Signal Transduct Target Ther* **2**, 17036 (2017).
30. Rojo de la Vega, M., Chapman, E. & Zhang, D.D. NRF2 and the Hallmarks of Cancer. *Cancer Cell* **34**, 21-43 (2018).
31. Fuchs, B.C. *et al.* Epithelial-to-mesenchymal transition and integrin-linked kinase mediate sensitivity to epidermal growth factor receptor inhibition in human hepatoma cells. *Cancer Res* **68**, 2391-2399 (2008).
32. Chong, J., Wishart, D.S. & Xia, J. Using MetaboAnalyst 4.0 for Comprehensive and Integrative Metabolomics Data Analysis. *Curr Protoc Bioinformatics* **68**, e86 (2019).
33. Xiang, Q. *et al.* Cabozantinib suppresses tumor growth and metastasis in hepatocellular carcinoma by a dual blockade of VEGFR2 and MET. *Clin Cancer Res* **20**, 2959-2970 (2014).
34. Kong, D. *et al.* Histone deacetylase inhibitors induce epithelial-to-mesenchymal transition in prostate cancer cells. *PLoS One* **7**, e45045 (2012).
35. Ji, M. *et al.* HDAC inhibitors induce epithelial-mesenchymal transition in colon carcinoma cells. *Oncol Rep* **33**, 2299-2308 (2015).

36. Padmanaban, V. *et al.* E-cadherin is required for metastasis in multiple models of breast cancer. *Nature* **573**, 439-444 (2019).
37. Bonzo, J.A., Ferry, C.H., Matsubara, T., Kim, J.H. & Gonzalez, F.J. Suppression of hepatocyte proliferation by hepatocyte nuclear factor 4alpha in adult mice. *J Biol Chem* **287**, 7345-7356 (2012).
38. Robert, K. *et al.* Cystathionine beta synthase deficiency promotes oxidative stress, fibrosis, and steatosis in mice liver. *Gastroenterology* **128**, 1405-1415 (2005).
39. Li, F. *et al.* Human betaine-homocysteine methyltransferase (BHMT) and BHMT2: common gene sequence variation and functional characterization. *Mol Genet Metab* **94**, 326-335 (2008).

REVIEWERS' COMMENTS:

Reviewer #1 (Remarks to the Author):

As acknowledged by the authors, there are still many questions that they have not been able to address. However, I think the study has been improved and will be of interest. I could support publication in Nature Comms at this point.

Reviewer #2 (Remarks to the Author):

In my opinion, this remains a strong paper on the mechanisms of how HNF4a regulates SAA pathways, but a weak paper concerning the implication that Met restriction could be therapeutically valuable in treating liver cancer. The new data provided on measured concentrations of SAAs in the media of the Met/Cys-depleted culture conditions are appreciated and, indeed, I fully accept the responses/ new data provided for all of my previous concerns except the in vivo models (previous comment 8), as detailed below.

I had expressed skepticism that the subtle differences in Met content between the Met-deficient and control chow could have a substantial effect on how much Met is available to the tumors, and I requested that levels of circulating SAAs be measured. In response, the authors indicate that their collaborators measure a ~2-fold decrease in circulating Met using these diets, and they now cite that work. However, for several reasons, I find this insufficient.

1) A 2-fold decrease in circulating Met is small, unlikely to impact cell physiology, and unlikely to exceed variance in Met concentrations that will occur with normal diurnal feeding cycles, fasting, etc.

2) This is "historical" data from a fundamentally different study, and might not reflect actual serum Met concentrations in the present study. There is no indication of normal levels or diurnal fluctuation. Moreover, differences in SAA metabolism by the microbiome between this study and the cited study could further influence this 2-fold difference.

3) The response does not provide a quantitative value for circulating Met concentrations. For the cell culture experiments, the authors now provide actual Met/cystine concentrations (0.12/0 μM), which are ~1000-fold below that in complete medium. This important data is appreciated, and helps me establish expectations for how much Met might be required by these cells, and at what concentration a true Met-insufficiency might occur. Where do the circulating levels of Met/cystine in the mouse models lie? Specifically, does the Met-diminished diet result in circulating Met concentrations in the sub- μM range?

4) I feel that, in the presentation, the authors are not transparent with the readers on this. Although the response to review indicates that the decrease in circulating Met is ~2-fold, the manuscript only says that Met is lower (not that it is only 2-fold lower), and it cites the collaborators' paper rather than providing concentrations and magnitude of difference.

I am not convinced that in vivo studies had a physiologically relevant difference in circulating Met. Moreover, I do not feel that the highly significant and potentially high-impact implication that Met restriction might be beneficial in combination therapies for HCC is supported or justified. I feel that the measured values for serum Met/cystine in the mice, including normal fluctuations/variance, should be presented and that there needs to be discussion of whether the diminished Met concentrations needed to impact these cells in culture could be safely achieved in vivo, or, if not, how much of a decrease in Met would need to be achieved in animals or patients to have an impact on cancer outcomes.

Reviewer #3 (Remarks to the Author):

The authors have carefully addressed the previous comments. The paper is now appropriate for publication.

Reviewer's #2 report:

In my opinion, this remains a strong paper on the mechanisms of how HNF4a regulates SAA pathways, but a weak paper concerning the implication that Met restriction could be therapeutically valuable in treating liver cancer. The new data provided on measured concentrations of SAAs in the media of the Met/Cys-depleted culture conditions are appreciated and, indeed, I fully accept the responses/ new data provided for all of my previous concerns except the in vivo models (previous comment 8), as detailed below.

I had expressed skepticism that the subtle differences in Met content between the Met-deficient and control chow could have a substantial effect on how much Met is available to the tumors, and I requested that levels of circulating SAAs be measured. In response, the authors indicate that their collaborators measure a ~2-fold decrease in circulating Met using these diets, and they now cite that work. However, for several reasons, I find this insufficient.

We would like to thank both reviewer #1 and #2 again for taking time to provide critical but constructive suggestions/comments to our studies. They are all important and value points.

Regarding the concerns raised by the reviewer #2, we have the following responses:

1) A 2-fold decrease in circulating Met is small, unlikely to impact cell physiology, and unlikely to exceed variance in Met concentrations that will occur with normal diurnal feeding cycles, fasting, etc.

Firstly, we would like to clarify that the 50% reduction of circulating Met after 2-day MCR diet feeding is the net reduction after controlling any possible diurnal feeding variables, as plasma was collected from CTR and MCR fed mice at a comparable Zeitgeber timeframe with the same fasting/fed condition.

Secondly and more importantly, a 2-fold decrease in circulating Met should not be considered small. Circulating metabolites are tightly controlled by various hormonal and environmental cues in vivo. "Small" alterations of many nutrients often lead to drastic impacts on physiology and pathology. For instance, a 50% increase in normal circulating glucose levels could lead to diabetes, whereas a 50% reduction of normal circulating glycose levels will result in hypoglycemia, despite that these changes may not exceed variance in blood glucose concentrations occurring with normal diurnal feeding cycles. As documented by extensive literature, in the case of methionine, the chronic impacts of this 50% reduction of circulating Met under MCR diet include alteration of methionine, cysteine, and glutathione metabolism in humans¹, and extension of life span, delay of aging, prevention of metabolic diseases, suppression of cancer growth, and sensitization of cancer cells to chemotherapy and radiation in mice¹⁻⁴.

2) This is "historical" data from a fundamentally different study, and might not reflect actual serum Met concentrations in the present study. There is no indication of normal levels or diurnal fluctuation. Moreover, differences in SAA metabolism by the microbiome between this study and the cited study could further influence this 2-fold difference.

The 50% reduction of plasma Met levels induced by an MCR diet compared to a chow diet that contains 0.4% Met¹ (Fig. 1d therein, also see attached Figure R1) was obtained in an another facility for a different study. However, the goal of this cited study by our co-author Jason

Locasale was to analyze the impact of dietary methionine restriction on tumor progression, which is “fundamentally” the same as the present study.

Moreover, results from independent studies have proven that the impact of MCR diet on circulating and tissue Met levels is independent of facilities and animal species despite possible impact of diurnal fluctuation. For example, compared to the Control diet that contains 0.86% Met, an MCR diet that contain 0.12% Met can reduce the plasma Met concentration by 70% (99.89 μM to 28.79 μM) in C57BL/6J mice, 30% (37.58 μM to 27.44 μM) in NSG mice bearing CRC119 tumors, and 40% (52.35 μM to 32.98 μM) in NSG mice bearing CRC240 tumors, in 3 weeks at Duke University facility ¹ (Extended Data Table 1 in ¹), or 80% (248 μM to 46.9 μM) in athymic nude mice in 12 weeks at Penn State University College of Medicine (Table 2 in ⁵). A similar 80% restriction of dietary Met also reduces plasma Met levels from 12.74 μM to 6.55 μM in 3 weeks in humans in Penn State University Clinical Research Center (Extended Data Table 1 in ¹). Therefore, regardless of facilities and animal species, 80% MCR diet can result in a 30%-80% reduction of circulating Met levels in the timeframe of days to weeks.

3) The response does not provide a quantitative value for circulating Met concentrations. For the cell culture experiments, the authors now provide actual Met/cystine concentrations (0.12/0 μM), which are ~1000-fold below that in complete medium. This important data is appreciated, and helps me establish expectations for how much Met might be required by these cells, and at what concentration a true Met-insufficiency might occur. Where do the circulating levels of Met/cystine in the mouse models lie? Specifically, does the Met-diminished diet result in circulating Met concentrations in the sub- μM range?

Direct translation of our in vitro results into in vivo application is not advised. Our current cell culture MCR condition was chosen to investigate the acute response of cultured cells to MCR-induced transcriptional response (AAR) and cell death, and a drastic restriction of Met/Cys helped to achieve our goals within a short experimental timeframe. However, this ~1000-fold reduction is not really necessary in vivo: (1) our data in Figure 3c indicate that a 10-20-fold reduction of methionine (from 130 μM to 13 or 6.5 μM Met) is sufficient to induce significant cell death in all three epithelial liver cancer cells in just 24 hours in culture; (2) in vivo MCR diet feeding is typically performed over weeks/months. This type of chronic treatment usually does not require a drastic reduction of Met levels. Specifically, the commonly used MCR diets that have 80-85% restriction of dietary Met are sufficient to reduce the circulating Met levels and chronically affect aging, metabolic diseases, and cancer growth and therapy ¹⁻⁴.

Importantly, we recently verified the therapeutic potential of our present MCR regiment (80% restriction of dietary methionine) in liver cancer treatment in a pilot experiment in collaboration with Dr. Xuemei Tong at Shanghai Jiao Tong University School of Medicine. As shown in new Supplementary Fig. 8, in a diethylnitrosamine (DEN)/high-fat-diet (HFD)-induced liver cancer model, 4-month of the MCR diet feeding is sufficient to significantly suppress liver cancer formation/growth in mice. Therefore, we are confident that the current MCR regiment could be potentially employed in combination therapies for liver cancer treatment.

For the specific question regarding the circulating levels of Met in mouse, based on our and other studies, the range is from 20-300 μM , and the MCR diet results in a 50% reduction of circulating Met within the first two days, and 0-70% reduction of circulating Met around 3 weeks, depending on mouse strains and experimental conditions ^{1,5}. Again, these acute and chronic

metabolic alterations have been shown to be sufficient to influence the tumor growth and therapeutic outcomes.

4) I feel that, in the presentation, the authors are not transparent with the readers on this. Although the response to review indicates that the decrease in circulating Met is ~2-fold, the manuscript only says that Met is lower (not that it is only 2-fold lower), and it cites the collaborators' paper rather than providing concentrations and magnitude of difference.

We apologize that our initial response was “not transparent”. Due to the variable nature of reported circulating Met levels in different mouse strains under different experimental conditions (from 20-300 μ M), as well as the slight difference between our MCR diet (0.172% Met) vs the MCR diet in cited studies (0.12% Met), we did not feel it is accurate to provide the concentrations and a magnitude of difference that cannot be generalized. However, per the request, we modified the sentence on Page 16 to the following: “Similar MCR diet has been shown to reduce plasma methionine levels by 50% within two days and alter methionine metabolism in colorectal PDX tumors and liver tissues in mice¹.”

I am not convinced that in vivo studies had a physiologically relevant difference in circulating Met. Moreover, I do not feel that the highly significant and potentially high-impact implication that Met restriction might be beneficial in combination therapies for HCC is supported or justified. I feel that the measured values for serum Met/cystine in the mice, including normal fluctuations/variance, should be presented and that there needs to be discussion of whether the diminished Met concentrations needed to impact these cells in culture could be safely achieved in vivo, or, if not, how much of a decrease in Met would need to be achieved in animals or patients to have an impact on cancer outcomes.

We apologize again that due to the current coronavirus pandemic, we are not able to quickly perform the requested experiments to measure circulating Met levels in our own mouse cohort. However, we feel that the reviewer's concerns have already been adequately addressed by the literature reports^{1, 5, 6} and our new data with the DEN/HFD liver cancer model (Supplementary Fig. 8), as indicated in our above responses.

To specifically address the reviewer's concern, we added the following paragraph in our Discussion on Page 24: “Our study has a technical limitation. Although we demonstrate that the MCR diet we used is able to suppress HNF4 α -positive liver cancer growth in mice using two independent liver cancer models (xenografted model in Fig. 6h and 6i, and DEN/HFD model in Supplementary Fig. 8), we did not directly measure the methionine levels in blood and tumors in our setup and confirm that the MCR diet can indeed reduce circulating and tissue methionine. On the other hand, a 30-80% reduction of circulating methionine levels has been previously observed in both mice and humans after days to weeks of 80% dietary methionine restriction from independent research facilities^{13, 15, 75}. Nevertheless, future analysis of circulating and tissue methionine concentrations during methionine-containing diet feeding as well as their variance during normal diurnal cycle will help to better assess the therapeutic importance of dietary methionine restriction in the treatment of human liver cancer patients.”

References:

1. Gao, X. *et al.* Dietary methionine influences therapy in mouse cancer models and alters human metabolism. *Nature* **572**, 397-401 (2019).

2. Orentreich, N., Matias, J.R., DeFelice, A. & Zimmerman, J.A. Low methionine ingestion by rats extends life span. *J Nutr* **123**, 269-274 (1993).
3. Miller, R.A. *et al.* Methionine-deficient diet extends mouse lifespan, slows immune and lens aging, alters glucose, T4, IGF-I and insulin levels, and increases hepatocyte MIF levels and stress resistance. *Aging Cell* **4**, 119-125 (2005).
4. Sanderson, S.M., Gao, X., Dai, Z. & Locasale, J.W. Methionine metabolism in health and cancer: a nexus of diet and precision medicine. *Nat Rev Cancer* **19**, 625-637 (2019).
5. Hens, J.R. *et al.* Methionine-restricted diet inhibits growth of MCF10AT1-derived mammary tumors by increasing cell cycle inhibitors in athymic nude mice. *BMC Cancer* **16**, 349 (2016).
6. Mentch, S.J. *et al.* Histone Methylation Dynamics and Gene Regulation Occur through the Sensing of One-Carbon Metabolism. *Cell Metab* **22**, 861-873 (2015).